# An Engineered *sgsh* Mutant Zebrafish Recapitulates Molecular and Behavioural Pathobiology of Sanfilippo Syndrome A/MPS IIIA

**DOI:** 10.3390/ijms22115948

**Published:** 2021-05-31

**Authors:** Alon M. Douek, Mitra Amiri Khabooshan, Jason Henry, Sebastian-Alexander Stamatis, Florian Kreuder, Georg Ramm, Minna-Liisa Änkö, Donald Wlodkowic, Jan Kaslin

**Affiliations:** 1Australian Regenerative Medicine Institute, Monash University, Clayton, VIC 3800, Australia; Alon.Douek@monash.edu (A.M.D.); Mitra.AmiriKhabooshan@monash.edu (M.A.K.); Sebastian-Alexander.Stamatis@monash.edu (S.-A.S.); Florian.Kreuder@monash.edu (F.K.); 2Neurotoxicology Lab, School of Science (Biosciences), RMIT University, Bundoora, VIC 3083, Australia; s3740720@student.rmit.edu.au (J.H.); Donald.Wlodkowic@rmit.edu.au (D.W.); 3Ramaciotti Centre for Cryo-Electron Microscopy, Monash University, Clayton, VIC 3800, Australia; Georg.Ramm@monash.edu; 4Department of Biochemistry and Molecular Biology, Biomedicine Discovery Institute, Monash University, Clayton, VIC 3800, Australia; 5Centre for Reproductive Health and Centre for Cancer Research, Hudson Institute of Medical Research, Clayton, VIC 3168, Australia; minni.anko@hudson.org.au; 6Department of Molecular and Translational Sciences, Monash University, Clayton, VIC 3800, Australia

**Keywords:** mucopolysaccharidosis, Sanfilippo syndrome, lysosomal storage disorder, childhood dementia, heparan sulfate, zebrafish, animal disease model, neuroinflammation, CRISPR/Cas9

## Abstract

Mucopolysaccharidosis IIIA (MPS IIIA, Sanfilippo syndrome type A), a paediatric neurological lysosomal storage disease, is caused by impaired function of the enzyme *N*-sulfoglucosamine sulfohydrolase (SGSH) resulting in impaired catabolism of heparan sulfate glycosaminoglycan (HS GAG) and its accumulation in tissues. MPS IIIA represents a significant proportion of childhood dementias. This condition generally leads to patient death in the teenage years, yet no effective therapy exists for MPS IIIA and a complete understanding of the mechanisms of MPS IIIA pathogenesis is lacking. Here, we employ targeted CRISPR/Cas9 mutagenesis to generate a model of MPS IIIA in the zebrafish, a model organism with strong genetic tractability and amenity for high-throughput screening. The *sgsh^Δex5−6^* zebrafish mutant exhibits a complete absence of Sgsh enzymatic activity, leading to progressive accumulation of HS degradation products with age. *sgsh^Δex5−6^* zebrafish faithfully recapitulate diverse CNS-specific features of MPS IIIA, including neuronal lysosomal overabundance, complex behavioural phenotypes, and profound, lifelong neuroinflammation. We further demonstrate that neuroinflammation in *sgsh^Δex5−6^* zebrafish is largely dependent on interleukin-1β and can be attenuated via the pharmacological inhibition of Caspase-1, which partially rescues behavioural abnormalities in *sgsh^Δex5−6^* mutant larvae in a context-dependent manner. We expect the *sgsh^Δex5−6^* zebrafish mutant to be a valuable resource in gaining a better understanding of MPS IIIA pathobiology towards the development of timely and effective therapeutic interventions.

## 1. Introduction

Mucopolysaccharidosis (MPS) IIIA, or Sanfilippo syndrome A, is a lysosomal storage disease that occurs due to loss-of-function mutations in the *SGSH* gene, whose product *N*-sulfoglucosamine sulfohydrolase (sulfamidase) is required for the degradation of the glycosaminoglycan (GAG) heparan sulfate (HS). HS in its proteoglycan form is abundant in diverse cells and tissues, reflecting its important roles in wide-ranging developmental and pathological contexts. HS can act as a ligand in Wnt signalling [1] and is an important mediator of developmental and physiological angiogenesis [2,3] and driver of tumour metastasis [4]. Indicative of this dual involvement in physiological as well as pathological cell behaviour, turnover of HS is equally important as its biosynthesis; thus, both processes must be tightly regulated. MPS IIIA initially presents in patients during early developmental stages as a failure to meet developmental milestones, coinciding with the onset of behavioural abnormalities including hyperactivity and impaired sleep patterns [5]. These clinical features precede a progressive decline in intellectual and cognitive functions [6,7,8], often leading to onset of dementia during early childhood. Mean age at death for MPS IIIA patients generally occurs between ~13–16 years of age across geographically distinct patient cohorts [9,10,11], with cause of death often relating to failure of autonomic functions (e.g., swallowing, breathing) leading to pneumonia or cardiorespiratory failure [12]. CNS dysfunction is profound, with patients exhibiting diverse features of CNS pathology including neuroinflammation, astrogliosis, and impaired neuronal autophagy [13], and is the basis for the bulk of morbidity associated with the condition. Conversely, somatic manifestations of MPS IIIA (e.g., craniofacial and skeletal dysmorphism, gastrointestinal dysfunction) are comparatively mild [14]. While much has been studied relating to the course of MPS IIIA progression and its clinical features, relatively little is known about the cellular mechanisms through which the disease manifests in patients. This represents a severe limitation for the development of disease-modifying therapeutic interventions. However, the most basal cellular dysfunction in MPS IIIA must relate to loss of SGSH enzyme activity and consequential accumulation of partially degraded HS. HS is degraded in a stepwise enzymatic process, with the majority of degradation occurring in the lysosome, in which constitutive components can be recycled into new GAGs. Specifically, SGSH removes the sulfate group from *N*-Sulfoglucosamine by hydrolysis to produce *D*-Glucosamine [15]; when SGSH activity is absent, subsequent steps in the HS degradation pathway (e.g., *NAGLU*-mediated acetylglucosaminidase activity impaired in MPS IIIB, *HGSNAT*-mediated acetyltransferase activity which is impaired in MPS IIIC, or *GNS*-mediated sulfatase activity in MPS IIID) cannot occur. Another subtype, putatively referred to as MPS IIIE, is thought to result from arylsulfatase G (*ARSG*) dysfunction, but this has only been observed in animal models [16,17] and has yet to be described in a human patient (most *ARSG* mutations are instead associated with Usher syndrome type 4 [OMIM 6181440]) [18,19,20,21]. MPS IIIA, the most common of the MPS III subtypes [22], tends to exhibit greater clinical severity and progresses more rapidly than other MPS III subtypes [10].

Due to its genetic tractability, high fecundity, ease of in vivo imaging, and advanced genetic toolbox, the zebrafish (*Danio rerio*) is an excellent model both for the study of fundamental developmental processes, as well as efficiently recapitulating many complex human diseases [23]. In the context of MPS IIIA, the use of the zebrafish as a disease model presents an opportunity to finely dissect the molecular and cellular mechanisms that underlie disease progression from its most fundamental causative events to late stage pathology. Zebrafish embryonic development occurs between fertilisation and hatching from the chorion at three days post-fertilisation (dpf), and is followed by a larval developmental period from four to seven dpf which is roughly comparable to the early postnatal period in mammals. During this period, neurodevelopment occurs rapidly, and is largely complete by five dpf. From seven dpf, zebrafish are considered juveniles until the onset of adulthood with acquisition of sexual maturity at approximately three months of age. These well-characterised development stages facilitate direct, time-resolved comparisons between the *sgsh^Δex5−6^* zebrafish model, other animal models of MPS IIIA, and MPS IIIA patients. To this end, we present the first zebrafish model of MPS IIIA, *sgsh^Δex5−6^*, which has been engineered to completely ablate Sgsh enzymatic activity and faithfully recapitulates the pathophysiology and progression of the disease in the context of the CNS. In lieu of examining features of MPS IIIA in senescent zebrafish, the purpose of this paper is to leverage the well-described utility of the zebrafish as a developmental model to identify early phenotypes and signs of pathobiology already evident during early developmental (embryonic or larval) stages. We expect the *sgsh^Δex5−6^* zebrafish to be a useful tool to better understand MPS IIIA, as well as other mucopolysaccharidoses, in order to aid in the development of future targeted therapies for MPS IIIA patients.

## 2. Results

### 2.1. Generation of the sgsh^Δex5−6^ Zebrafish Mutant by CRISPR/Cas9-Mediated Dual-gRNA Targeted Mutagenesis

The majority of MPS IIIA disease-causing alleles are hypomorphic in nature (OMIM 252900) and severity of disease correlates with degree of loss of SGSH enzyme activity [24]. We thus reasoned that in order to most comprehensively model MPS IIIA in the zebrafish, any mutant strain generated should exhibit little-to-no Sgsh enzyme activity. To accomplish this, a dual-gRNA CRISPR/Cas9-mediated genome editing approach was employed to delete a conserved part of the *sgsh* gene and generate a stable *sgsh* loss-of-function zebrafish mutant. To this end, we targeted two genomic sequences in exons 5 and 6 of zebrafish *sgsh* (Ensembl GRCz11 ENSDARG00000032087), which encode highly conserved components of the sulfatase domain of Sgsh protein (Figure 1A). This was predicted to result in an excision between two exon splice sites leading to splicing of exon 4 to exon 7 in mature *sgsh^Δex5−6^* mRNA. As exons 4 and 7 are not phase-compatible (i.e., exon 4 ends in reading frame position 0, while exon 7 starts in reading frame position 1), we did not expect to preserve translation through to the original termination codon, as a clean splicing event between exons 4 and 7 should result in a premature termination codon (PTC) early in exon 7. However, as exon 7 is the final exon of zebrafish *sgsh*, PTCs in this exon are insensitive to induction of nonsense-mediated mRNA decay (NMD) as all exon junction complexes have been displaced during translation. Thus, we expected the *sgsh^Δex5−6^* mutation to result in a truncated but stable mutant transcript, avoiding potential transcriptional adaptive responses which have been reported to attenuate some mutant phenotypes as a result of NMD [25].

An allele possessing the expected ~350 bp deletion as determined by PCR of genomic DNA was recovered from a founder. Sanger sequencing of PCR amplicons amplified from genomic DNA from homozygous *sgsh^Δex5−6^* mutants using primers flanking the edited region demonstrated the presence of a 357 bp deletion in the mutant allele (Figure 1B). This comprised a deletion of the region between both gRNA-targeting sequences, with small fragments of the intervening region present in the genomic lesion (Appendix A). To characterise the composition of *sgsh^Δex5−6^* mRNA, total RNA was extracted from maternal-zygotic F3 *sgsh^Δex5−6^* or wild type embryos and reverse-transcribed to cDNA. Semi-quantitative RT-PCR using a forward primer targeting a sequence in exon 1 and a reverse primer targeting exon 7 of *sgsh* demonstrated the presence of a truncated transcript that was expressed at levels comparable to *sgsh* transcript in wild type fish (Figure 1C). Sanger sequencing of the RT-PCR amplicon confirmed that the mutant transcript exhibited splicing of exons 4 and 7 (Appendix A), evidencing exon-skipping of the residual exon 5 fragment present in *sgsh^Δex5−6^* genomic DNA during pre-mRNA processing (Appendix A).

As the sulfatase domain of zebrafish Sgsh is encoded by *sgsh* exons 1–7 (Figure 1A), we sought to validate whether the deletion in the *sgsh^Δex5−6^* allele would be sufficient to completely ablate the sulfatase activity of the enzyme. First, the amino acid sequence encoded by the *sgsh^Δex5−6^* allele was inferred from Sanger sequencing of the RT-PCR product (Appendix A), which demonstrated a PTC 81 bp into exon 7. Next, SWISS-MODEL [26] was used to model in silico the structure of Sgsh^Δex5−6^ compared to the wild type homodimeric form of the protein, using the crystal structure of human SGSH (PDB: 4MHX) [27] as a template. The predicted structure of the wild type homodimer (Appendix A) closely resembled the structure of human SGSH [27]. By contrast, the 256-amino acid mutant protein lacks several conserved residues present at the dimer interface [27] and thus could not be modelled as a dimer (Appendix A), suggesting non-covalent homodimerisation of the mutant protein does not occur in the *sgsh^Δex5−6^* zebrafish. Additionally, while the highly-conserved sulfatase active site motif C(X)PSR [27] was present in both wild type and mutant amino acid sequences, three conserved residues present at the active site (D282, N283 and R291) are absent in the mutant protein product. Taken together, these data suggest enzymatic function would be ablated in the *sgsh^Δex5−6^* zebrafish mutant. As such, we sought to directly experimentally evaluate the degree of sulfatase activity exerted by mutant Sgsh^Δex5−6^ enzyme. 

### 2.2. sgsh^Δex5−6^ Mutant Zebrafish Exhibit Total Absence of Sgsh Activity and Accumulate Heparan Sulfate Partial Degradation Products

To assess the degree of enzymatic function impaired as a consequence of the *sgsh^Δex5−6^* mutation, total protein derived from wild type, heterozygous, and homozygous *sgsh^Δex5−6^* embryos was assayed using a protocol modified from Karpova et al. [28] and Whyte et al. [29]. Briefly, this two-step enzymatic assay first requires desulfation of a 4-methylumbelliferone (4MU)-conjugated glucosamine substrate (4MU-αGlcNS) by Sgsh in total protein, followed by hydrolysis of the fluorescent 4MU from the desulfated glucosamine (4MU-αGlcNH2) by supplemented bacterial α-glucosidase (Figure 1D). Relative to wild type-derived total protein, protein derived from *sgsh^Δex5−6^* F3 maternal-zygotic homozygous embryos at 24 h post-fertilisation (hpf) displayed a complete absence of Sgsh activity, while heterozygous *sgsh^Δex5−6^* embryos exhibited a ~50% reduction in Sgsh activity (Figure 1E); this was in line with a genetic dose-dependent reduction in functional enzyme abundance. Liver (Appendix A) and brain (Figure 1E) from six-month-old adult fish of respective genotypes were assayed for Sgsh enzymatic activity. In agreement with the embryonic data, the *sgsh^Δex5−6^* mutant tissues were completely lacking in Sgsh enzymatic activity (Figure 1F).

Furthermore, accumulation of HS GAGs in tissues over time is a hallmark of MPS IIIA directly resulting from Sgsh enzyme dysfunction. To quantitate HS GAG accumulation in the *sgsh^Δex5−6^* zebrafish, we performed liquid chromatography-tandem mass spectrometry (LC-MS/MS) according to He et al. [30] on tissue lysates derived from whole embryos (24 hpf), or young (six months old) and aged (18 months old) adult brains from either genotype. In line with *sgsh^Δex5−6^* mutants showing no Sgsh enzymatic activity, a significant increase in HS content was detected in mutants in all groups assayed. In 24 hpf embryos, the derived HS disaccharide (i.e., dibutylated GlcN-GlcUA) levels were 2.1× higher in the *sgsh^Δex5−6^* mutants compared to wild types (Figure 1G, p = 0.0021). In young adult brains, HS levels were increased 22.5-fold in mutants compared to wild types (Figure 1H, *p* = 0.0171), while in aged brains a 49.9-fold increase in HS was observed in mutant tissue compared to age-matched wild types (Figure 1H, p = 0.0031). Taken together, these data demonstrate a complete loss of Sgsh enzyme activity and a consequent massive accumulation of HS over time in the CNS of *sgsh^Δex5−6^* homozygous zebrafish, with an absence of meaningful compensatory mechanisms in HS catabolism.

### 2.3. Behavioural Perturbations and Cellular Pathology in the CNS of Homozygous sgsh^Δex5−6^ Zebrafish Recapitulates Features of Human MPS IIIA

As hyperactivity and other abnormal behaviours are characteristic of early stages of MPS IIIA pathology, wild type and *sgsh^Δex5−6^* homozygous larvae were assayed at 5 dpf (roughly comparable to early childhood in humans) for activity using a custom, video-based behavioural tracking system [31]. Neural development occurs rapidly in zebrafish and is largely complete by 5 dpf; this is therefore an appropriate time point at which to assay early-life behaviours. Compared to wild type siblings, *sgsh^Δex5−6^* larvae broadly displayed generalised hyperactivity upon tracking of basal movement. While cumulative distance travelled was not significantly altered between genotypes (Figure 2A, *p* = 0.5544), measures of negative thigmotaxis (time spent away from a wall, a readout of reduced anxiety) indicated highly increased hyperactive behaviour including a significantly reduced wall preference ratio (T_w_, Figure 2B, *p* < 0.0001) and highly increased frequency of exploratory zone intercepts (Figure 2C,D, *p* < 0.0001). The normal disease course of MPS IIIA in patients follows a loss of this early hyperactive state as functional neurodegeneration proceeds. To examine if this was phenocopied in the *sgsh^Δex5−6^* mutant, we analysed swimming behaviour in adult wild type and homozygous *sgsh^Δex5−6^* fish. During behavioural assays, adult zebrafish generally exert thigmotactic behaviours that manifest as a positional preference for the walls of their tank [32]. By defining a central zone in the arena (the ‘exploratory zone’) to represent an inverse function of wall preference, anxiety-related exploratory behaviour in adult zebrafish was quantified in terms of number of zone intercepts. Relative to age-matched 6-month-old wild type adult zebrafish, *sgsh^Δex5−6^* homozygous adults displayed significant decreases in cumulative distance travelled (Figure 2E, *p* < 0.0001) and average movement velocity (Figure 2F, *p* < 0.0001), reflecting an inversion of the larval hyperactivity phenotype. Furthermore, and despite their reduction in overall movement, *sgsh^Δex5−6^* adult zebrafish entered the exploratory zone more than wild type fish (Figure 2H, *p* = 0.0146) and spent a significantly increased time within the exploratory zone (Figure 2G, *p* = 0.0029). MPS IIIA patients exhibit a loss of fear associated with progressive amygdala atrophy [33]. Thus, impaired thigmotaxis in combination with reduced mobility in adult *sgsh^Δex5−6^* homozygotes compared to wild type controls may indicate the onset of cognitive deficit. Collectively, these data indicate the presence of an early-developmental hyperactivity phenotype followed by an adult hypoactivity phenotype with features of cognitive decline, reflecting behavioural features of disease progression in both MPS IIIA patients [14] and MPS IIIA murine models [34].

Maternal-zygotic *sgsh^Δex5−6^* mutants were viable well into adulthood and fertile, and did not display any obvious embryonic nor adult gross morphological phenotype. However, CNS rather than somatic pathology is the most significant source of morbidity in MPS IIIA. In order to assess the presence of CNS-related pathological features in homozygous *sgsh^Δex5−6^* zebrafish, we evaluated brains from young and aged *sgsh^Δex5−6^* homozygotes for cellular features of MPS IIIA. First, we employed the LysoTracker vital dye, which is specific to acidic subcellular compartments and can be used to detect both lysosomes and autophagic organelles such as the autolysosome [35]. Live-imaging of the telencephalon of 7 dpf larvae stained with the LysoTracker Red DND-99 probe revealed a significantly increased abundance of small intracellular LysoTracker-labelled vesicles in homozygous *sgsh^Δex5−6^* mutants relative to wild type larvae (Figure 3A–C, arrowheads, 5.51-fold increase, *p* < 0.0001). In addition, large LysoTracker-positive autophagosomes were consistently observed in significantly higher numbers in *sgsh^Δex5−6^* mutants than in wild types (Figure 3A,B,D, 3.63-fold increase, *p* = 0.0001, arrows). Further, ultrastructural analysis by transmission electron microscopy (TEM) of different regions of both wild type (Figure 3E–G) and homozygous *sgsh^Δex5−6^* (Figure 3H–J) adult brains demonstrated increased abundance of distended lysosomes, signifying the presence of lysosomal pathology in both the larval and adult CNS. TEM analysis also confirmed the presence of autophagosomes in homozygous *sgsh^Δex5−6^* adult brain tissue, while these were not observed in adult wild type control tissue (n = 3 brains per each genotype, Figure 3I–J). Intriguingly, these autophagosomes were most frequently observed within axons and often contained mitochondria, suggesting the presence of elevated generalised neuronal autophagy as well as a possible axonal dystrophic phenotype in the *sgsh^Δex5−6^* mutant zebrafish.

Progressive cortical grey matter atrophy is frequently observed in MPS IIIA patients [36], and increased expression of apoptosis-related genes have been detected in regions of the brain of an MPS IIIA mouse model [37]. This led us to speculate whether an apoptotic phenotype was detectable in the CNS of the *sgsh^Δex5−6^* mutant at varying stages of life. To address this, *sgsh^Δex5−6^* homozygotes were crossed into the Tg(-3.5*ubb*:SECHsa.ANXA5-mVenus)^mq8^ transgenic line, which labels apoptotic cells in vivo [38]. Homozygous *sgsh^Δex5−6^* and wild type progeny were raised to 7 dpf, and larval brains were dissected and confocally imaged as whole-mounts for ANXA5-mVenus-expressing apoptotic bodies. Homozygous *sgsh^Δex5−6^* mutants exhibited a ~9-fold increase in the number of ANXA5-mVenus^+^ objects in the larval brain compared to wild types (Figure 3K–L, *p* = 0.0003). Analysis of apoptosis in adult brains using the ANXA5-mVenus transgenic line is complicated by accumulation of mVenus signal within both apoptotic cells and myelin sheaths (Appendix A). The latter occurs due to an abundance of the phospholipid phosphatidylserine (a target of secreted ANXA5) within myelin [39], and given that myelin is far more abundant a source of phosphatidylserine than apoptotic cells in the adult zebrafish CNS, the mVenus signal of apoptotic cells is heavily masked by that of myelin. To bypass this technical limitation, we employed a highly sensitive luminescent assay of Caspase-3/7 activity to quantify relative apoptosis in dissociated cells from the adult zebrafish brain of both wild types and *sgsh^Δex5−6^* homozygotes. In contrast to the apoptotic phenotype observed in larval *sgsh^Δex5−6^* brains as detected by the ANXA5-mVenus transgene, Caspase-3/7 activity was not significantly changed between *sgsh^Δex5−6^* adult CNS cells and cells from age-matched wild type controls (Appendix A, *p* = 0.9602, 30 min of reaction incubation). However, ultrastructural examination by TEM of brains from adult homozygous *sgsh^Δex5−6^* mutants revealed sparse numbers of large, electron-dense apoptotic-like bodies sporadically distributed across the brain (Appendix A). These bodies were associated with microglia with an activated, amoeboid morphology (Appendix A), and were not detected in CNS tissue from age-matched wild type siblings (n = 3 brains per each genotype). Taken together, CNS apoptosis is prevalent in early developmental stages in the *sgsh^Δex5−6^* mutant; however, a caspase-dependent apoptotic phenotype does not appear to be a major contributor to the disease state in the adult brain. Collectively, these data demonstrate that *sgsh^Δex5−6^* zebrafish display characteristic pathological features of MPS IIIA, highlighting its utility as an experimental model for the disease.

### 2.4. Early-Onset and Lifelong Neuroinflammation Is a Hallmark of sgsh^Δex5−6^ Zebrafish Mutant Pathology

Neuroinflammation is a hallmark of many neurological diseases and has previously been reported in various models of Sanfilippo syndrome subtypes, including MPS IIIA [37,40,41], IIIB [42], and IIIC [43]. This is generally recognised in the context of the mammalian CNS to refer to complementary activation of both microglial cells (the resident macrophage-like population of the CNS) and parenchymal astrocytes. While astrocyte reactivity has been demonstrated in the CNS of MPS IIIA mouse models [41,44], zebrafish do not possess parenchymal astrocytes, with immunoreactivity or transgene expression of GFAP (a widely used marker for reactive astrocyte identity in mammalian tissue) restricted to ventricular radial glial-like cells [45,46]. Furthermore, examination of microglial activation is complicated in the zebrafish by a lack of conserved markers of this cell state between mammalian and teleost contexts. Commonly used markers of activated microglia such as IB4 and Iba1 do not sensitively or specifically label these populations in zebrafish [47]; indeed, a sensitive and specific molecular marker for activated microglia in zebrafish is presently unknown [48]. To assess the presence and temporal dynamics of a neuroinflammatory phenotype in *sgsh^Δex5−6^* zebrafish, we first performed immunostaining on larval brains with the pan-leukocyte marker L-plastin (Lcp1) to detect microglia [49,50]. We opted to use L-plastin immunoreactivity as a readout of microglial activation, as in the brain it is a selective marker for microglial identity, and its membrane/cytoplasmic localisation has been shown to robustly report microglial activation on the basis of altered cellular morphology [49,51,52]. In a normal physiological state, microglia predominantly possess elaborate, ramified morphologies; contrastingly, under adverse conditions (e.g., CNS infection or traumatic injury) activated microglia display a condensed, amoeboid morphology [53,54]. At 7 dpf, wild type larval brains exhibited predominantly ramified microglia (Figure 4A′) distributed throughout the parenchyma, with comparatively few microglia exhibiting an activated amoeboid morphology (Figure 4A,C, 13.7-fold relative increase in ramified vs. amoeboid cells, *p* < 0.0001). In contrast, homozygous *sgsh^Δex5−6^* larval brains showed abundant activated microglia (Figure 4B′) relative to wild type controls (Figure 4B,C, *p* < 0.0001), but a heavily reduced number of ramified microglia (Figure 4B,C, 19.5-fold relative increase in amoeboid vs. ramified cells, *p* < 0.0001). Together, these characteristics indicate profound microglial activation and neuroinflammation in larval homozygous *sgsh^Δex5−6^* brains.

In order to determine whether neuroinflammation in *sgsh^Δex5−6^* homozygotes was restricted to developmental stages, or rather represented a chronic pathological feature of the *sgsh^Δex5−6^* CNS, L-plastin immunostaining was conducted on adult brain sections from *sgsh^Δex5−6^* homozygotes and age-matched wild type controls, and quantification was performed on midbrain (optic tectum and diencephalon) sections from either genotype due to the highly stereotyped morphology of this region. In wild type midbrain sections, most microglia displayed a ramified morphology (Figure 4D′) and were distributed predominantly throughout the parenchyma (Figure 4D,F, 7.3-fold relative increase in ramified vs. amoeboid cells, *p* < 0.0001). Homozygous *sgsh^Δex5−6^* midbrain sections displayed a heavily reduced number of ramified microglia relative to wild type controls (Figure 4E,F, 3.2-fold decrease in relative number of ramified cells in *sgsh^Δex5−6^* sections vs. wild type, *p* < 0.0001), but instead possessed significantly increased levels of ventricular surface-associated activated microglia (Figure 4E′) compared to controls (Figure 4E,F, 11.7-fold relative increase in amoeboid cells in *sgsh^Δex5−6^* sections vs. wild type, *p* < 0.0001). Furthermore, TEM of adult homozygous *sgsh^Δex5−6^* brains revealed the presence of large, electron-lucent intracellular vacuoles within perivascular microglia with activated morphology (Figure 4G–I). Such large intracellular vacuoles are characteristic of foamy vacuole-containing macrophages observed in several lysosomal storage disorders [55,56,57,58,59]. These data demonstrate that early-developmental neuroinflammation and microglial activation in homozygous *sgsh^Δex5−6^* zebrafish persists throughout life.

### 2.5. Pharmacological Blockade of Il-1β Maturation Ameliorates Early-Developmental Microglial Activation in sgsh^Δex5−6^ Larvae

Neuroinflammation is known to occur in and drive diverse lysosomal storage disorders, including the gangliosidosis Sandhoff disease [60], the demyelinating pathology Krabbe’s disease/globoid-cell leukodystrophy [61], and many subtypes of mucopolysaccharidoses including MPS IIIA [62,63], IIIB [64], and IIIC [43]. Indeed, gene therapy-driven mitigation of interleukin (Il)-1β-dependent neuroinflammation demonstrates its relevance as a driver of MPS IIIA by improving cognitive function in a murine model [62]. In order to determine whether the neuroinflammatory phenotype observed in the *sgsh^Δex5−6^* zebrafish MPS IIIA model involved Il-1β signalling, we investigated the effect of chronic pharmacological suppression of inflammatory signals on microglial activation of in *sgsh^Δex5−6^* mutants by early-life exposure to Ac-YVAD-cmk (YVAD), a selective, irreversible inhibitor of caspase-1. YVAD prevents the maturation of pro-Il-1β, the precursor of the inflammatory cytokine Il-1β, which in turn prevents its release from pro-inflammatory microglia.

Continuous treatment of wild type larvae with 50 µM YVAD from 3 to 7 dpf significantly reduced the number of ramified microglia in 7-dpf larval brains (Figure 5A,C,E, 1.5-fold decrease in ramified cells upon YVAD treatment in wild type larvae, *p* < 0.0001), but did not affect the already-low number of activated microglia compared to 0.1% DMSO-treated wild type controls (Figure 5A,C,E, *p* > 0.9999). Conversely, YVAD treatment reversed microglial activation in *sgsh^Δex5−6^* homozygotes, with a significant increase in the number of ramified microglia (7.8-fold increase in number of ramified cells upon YVAD treatment in *sgsh^Δex5−6^* larvae, *p* < 0.0001) as well as a significant decrease in the number of activated microglia (11.2-fold decrease in number of amoeboid cells upon YVAD treatment in *sgsh^Δex5−6^* larvae, *p* < 0.0001) upon treatment from 3–7 dpf with 50 µM YVAD, compared to 0.1% DMSO-treated homozygous *sgsh^Δex5−6^* controls (Figure 5B,D,E). Taken together, these data confirm the presence of a chronic neuroinflammatory state in *sgsh^Δex5−6^* homozygotes that is partially corrected upon inhibition of Il-1β signalling.

While we previously demonstrated the presence of early-life hyperactivity and reduced anxiety by way of a reduced wall preference in the larval and adult *sgsh^Δex5−6^* mutant (Figure 2), we viewed it necessary to determine whether this phenotype could be attenuated by neuroinflammatory rescue. To elucidate the broader phenotypic consequences of amelioration of neuroinflammation in *sgsh^Δex5−6^* larvae, we applied an escape-response test that assessed thigmotactic behaviours under either standard or ‘threatened’ environmental conditions (changes in ambient lighting mimicking the presence of a predator) [31]. In contrast to previous assays where basal activity (i.e., under unstimulated conditions) was examined, a different approach utilising a simulated adverse stimulus is necessary. To effectively assay anxiety-related behaviour, an anxiogenic stimulus is useful to resolve cause and effect, and sudden environmental changes in light levels have been shown to effectively and predictably induce stress responses [65,66]. Basal activity under constant illumination was not significantly altered by chronic treatment of larvae from 3 to 7 dpf with either 50 µM YVAD or 0.1% DMSO (Figure 5F—dampened hyperactivity between 5 dpf and 7 dpf cohorts, c.f. Figure 2, likely results from depletion of yolk-derived nutrients in the latter time point; see Methods). YVAD treatment did not alter overt thigmotactic behaviour under either standard or threatened conditions in wild type larvae compared to DMSO-treated wild type controls (Figure 5G); however, chronic treatment of *sgsh^Δex5−6^* larvae with YVAD increased the amount of thigmotactic behaviour to levels nearer to, but slightly less than, wild type larvae under non-threatened conditions (Figure 5G, two-way ANOVA *p* < 0.0001 for variation in data due to treatment condition). In contrast to non-threatened conditions, impaired thigmotaxis seen in *sgsh^Δex5−6^* larvae under threatened conditions was refractory to YVAD treatment (Figure 5G). These data reflect the complexity of behavioural phenotypes observed in the MPS IIIA-like state of the *sgsh^Δex5−6^* mutant, and indicate that neuroinflammation is an important contributing factor to context-specific aspects of the observed behavioural abnormalities.

## 3. Discussion

In recent years, significant progress has been made towards better understanding the factors underlying and driving the mucopolysaccharidoses, including MPS IIIA. Despite these advances, an effective treatment paradigm remains elusive, partially due to the phenotypic complexity of MPS IIIA, as well as the still indeterminate mechanistic explanation of the link between the aetiology of the disease (i.e., impaired HS catabolism) and the manifestation of severe, progressive CNS dysfunction. As a model organism, the zebrafish has been used for diverse purposes ranging from ecotoxicology [67], to evolutionary [68] and developmental biology [69], to myriad uses in the biomedical sciences including regeneration of the CNS [70,71] and skeletal muscle following profound traumatic injury [72]. Most pertinent to this study, however, is the utility of the zebrafish as a model for inherited genetic disease [23]. With its excellent genetic tractability and strong genomic conservation relative to humans, as well as high fecundity and rapid ex utero development, the zebrafish represents a prime candidate for the modelling of MPS IIIA. Such a model would provide a resource for both in-depth investigation of the fundamental processes underlying MPS IIIA, as well as a robust experimental context for the high-throughput testing of candidate therapies for rapid clinical translation.

To this end, we present in this study the *sgsh^Δex5−6^* zebrafish, a rationally-engineered *sgsh* mutant designed to recapitulate MPS IIIA at its most severe with the aim of providing a tool to facilitate in-depth study of MPS IIIA pathobiology in ways not feasible with other MPS IIIA model organisms. In designing the *sgsh^Δex5−6^* mutant, we took into consideration recently described transcriptional adaptive compensatory responses observed in diverse mutants bearing a small indel or point mutation leading to an in-frame premature termination codon (PTC) [25,73]. Such deletions may lead to nonsense-mediated decay (NMD)-based degradation of the mutant mRNA, which can subsequently induce upregulation of genes with sequence similarity (and possible compensatory function), thereby mitigating or confounding downstream phenotypes [25,73]. Coupled with an appreciation of which exons encode the sulfatase domain of Sgsh protein, simultaneous targeting of Cas9 to two sequences located between a pair of intact exon splice donor/acceptor sites facilitates the derivation of the *sgsh^Δex5−6^* allele. While the *sgsh^Δex5−6^* allele results in a PTC due to a frameshift resulting from the incompatibility of donor and acceptor splice site phases for exons 4 and 7, respectively, this PTC occurs in the final exon and renders the Sgsh^Δex5−6^ gene product insensitive to NMD as all exon junction complexes present in the mutant mRNA (including at the junction between exons 4 and 7) are displaced. As such, this allele encodes a transcriptionally stable but completely non-functional Sgsh protein. With this, subsequent modelling of MPS IIIA can be performed with confidence that cryptic transcriptional adaptive responses to the presence of the mutant allele do not occur through any currently known mechanism.

In addition to the *sgsh^Δex5−6^* mutant, several animal models of MPS IIIA exist—both spontaneously occurring and engineered. These include spontaneous canine mutants [74,75], the well-characterised spontaneous murine D31N mutant [76], a conditional *Sgsh* mouse mutant [77], and *Drosophila* knock-down models of MPS IIIA [78]. In variable manners, all of these models recapitulate pathological features of human MPS IIIA. Here, we demonstrate that the cardinal features of MPS IIIA are similarly recapitulated in the *sgsh^Δex5−6^* zebrafish, which displays a total absence of Sgsh activity leading to systemic heparan sulfate accumulation, lysosomal distention and overabundance, increased CNS autophagy, complex behavioural phenotypes and chronic neuroinflammation. Previous animal models of MPS IIIA have displayed behavioural phenotypes specific to the model. For example, congenic B6.Cg-*Sgsh*^mps3a^ mice (~3% residual enzymatic activity relative to unaffected controls) exhibit hyperactivity [34], impaired learning behaviour upon subjection to the Morris water maze [79], and progressively acquire autistic-like behaviours with age [80]. Multiple studies have demonstrated early hyperactivity in MPS IIIA mice and/or later hypoactivity [79,81,82,83,84,85] (with some apparently sex-dependent disparate results [34]). Similarly, an RNAi-mediated *Drosophila* knockdown model of MPS IIIA displayed impaired climbing ability as a function of progressive locomotor deficiency [78]. Using the novel *sgsh^Δex5−6^* zebrafish model, we demonstrate the presence of comparable complex, progressive behavioural phenotypes that display phenotypic inversion between larval and adult stages, resembling the progression of activity-related behavioural symptoms in both MPS IIIA patients [33] and animal models. Throughout the course of our analyses, we did not observe a noticeable difference in survival between *sgsh^Δex5−6^* homozygous mutants and wild type siblings, nor did we observe catastrophic loss of motor function in aged mutant zebrafish. This is striking given the devastating nature of late-stage pathology in MPS IIIA patients with severe forms of the disease. At present, we are unable to specifically determine the reason(s) for greater tolerability of the MPS IIIA-like disease state in zebrafish. However, the robustness of this model, coupled with the many conserved cellular features of pathology described here, provides an opportunity for researchers to examine disease progression over lengthy experimental periods without significant attrition of samples due to disease morbidity.

Fluorometric assaying of Sgsh enzymatic activity demonstrated that irrespective of tissue analysed, *sgsh^Δex5−6^* homozygotes exhibit a complete loss of enzyme activity. This cannot be attributed to failure of our total protein isolation protocol to capture the resulting truncated mutant protein, as both wild type (~58 kDa) and predicted mutant (~29 kDa) protein products are well above the 10 kDa molecular weight cut-off (MWCO) in the method used for purification. We interpret the discrepant fluorescence values between adult liver and brain samples as a function of the relative abundance of Sgsh enzyme in total protein >10 kDa in each tissue type, with Sgsh exhibiting greater relative abundance among proteins >10 kDa in liver compared to brain. This finding is notable as it is not concordant with the distribution of uronic acid (UA, representative of total GAG burden) across multiple tissues in an MPS IIIA mouse model, where GAGs were disproportionately abundant in kidney, while not significantly different in abundance between brain and liver [86]. Given that many cases of MPS IIIA result from hypomorphic alleles exhibiting minimal but not completely abolished Sgsh enzyme activity, it remains unclear whether relative native abundance of Sgsh protein across different tissues is clinically relevant in determining involvement of a given tissue or organ in MPS IIIA pathogenesis. Future experiments with the *sgsh^Δex5−6^* zebrafish will seek to determine whether tissue-specific Sgsh protein and GAG abundance is similar to that of MPS IIIA mice, thereby facilitating better cross-model phenotypic comparisons. Additionally, future experiments will be conducted to examine directly the activities of other lysosomal enzymes (e.g., those involved in ganglioside degradation, such as *hexa* and *hexb*) to establish whether features of HS catabolism other than *Sgsh* activity are also perturbed in the *sgsh^Δex5−6^* zebrafish mutant.

Highly-sensitive quantification by LC-MS/MS of HS GAG in *sgsh^Δex5−6^* mutants and wild type siblings demonstrates the direct consequence of total loss of *Sgsh* enzymatic function. While examination of HS disaccharide abundance between young (6-month-old) and aged (18-month-old) *sgsh^Δex5−6^* zebrafish brains demonstrates a massively increased HS GAG burden over time in the CNS, we were able to resolve significantly increased systemic HS accumulation well prior to when the onset of CNS dysfunction would be expected. As early as 24 hpf, *sgsh^Δex5−6^* homozygotes display roughly twice the amount of systemic HS GAG relative to wild type siblings. This finding clearly demonstrates that significantly increased levels of HS are present in the *sgsh^Δex5−6^* mutant well prior to the onset of cellular dysfunction leading to an MPS IIIA-like pathological state. Heparan sulfate is involved in guiding diverse developmental processes due to its capacity to bind to a wide variety of proteins [87]; for example, HS is known to bind directly to Fgf8 [88,89] and directs formation of Fgf8 morphogen gradients in early vertebrate embryos via regulation of a source-sink mechanism [90]. Indeed, abolition of HS by heparanase I treatment significantly shifts the dynamics of gradient formation by altering the diffusibility of the morphogen across developmental axes [90]. It is presently unknown if such developmental processes are perturbed during early embryonic development of MPS IIIA patients. Studying this is complicated by a lack of routine screening for MPS III, and an absence of overt somatic disease coupled with CNS symptom-driven clinical presentation almost exclusively well after the postnatal period. Prenatal diagnosis of MPS III subtypes may be achieved by enzymatically assaying samples derived by chorionic villus sampling or amniocentesis, but this is unlikely to be undertaken in the absence of family history of Sanfilippo syndrome given the absence of overt signs of embryonic pathology in the gestational period [91]. Despite this, further study using experimental models of MPS III, such as the *sgsh^Δex5−6^* zebrafish mutant, into the presence of early developmental abnormalities resulting from overabundance of HS is warranted, as this may provide clues as to (i) the disproportionate symptomatic burden of the disease in the CNS relative to other tissues, (ii) whether early developmental HS overabundance can ‘prime’ patients into a proto-pathological state, and (iii) whether therapeutic interventions towards systemic GAG reduction performed during embryonic development may delay the onset, or mitigate the severity of MPS III in the postnatal period.

Examination of the brain in *sgsh^Δex5−6^* mutant larvae demonstrates that onset of cellular features of CNS pathology is prominent during larval stages of development. Using the telencephalon of 7 dpf larvae as a representative structure, LysoTracker staining of acidic subcellular compartments demonstrated a marked increase in lysosomal abundance within the grey matter of the telencephalon. This reflects findings previously described in *Drosophila* models of MPS IIIA [78]. Increased levels of otherwise rare CNS autophagosomes were detected in the *sgsh^Δex5−6^* mutant using this method. Furthermore, apoptotic cells (as detected by a ubiquitously-expressed, secreted ANXA5-mVenus transgene) are significantly increased in 7 dpf larval *sgsh^Δex5−6^* brains. Collectively, these features are highly suggestive of cellular stress in the CNS during early development, but alone do not explain the nature of MPS IIIA disease progression. Ultrastructural examination of CNS tissue from aged adult *sgsh^Δex5−6^* homozygotes and wild-type siblings confirmed that these features were sustained beyond larval developmental stages. In *sgsh^Δex5−6^* homozygotes, axons frequently displayed an increased abundance of lysosomes, and occasionally possessed large autophagic bodies often containing mitochondria. Apoptotic cells with close association to activated microglia were also identified in the homozygous *sgsh^Δex5−6^* CNS, though the identity of these cells could not be determined due to the hyper-dense appearance of apoptotic cells in routine TEM preparations. Collectively, these features were rarely or never observed during ultrastructural examination of CNS tissue from age-matched wild type siblings. However, both a lack of change in brain-wide Caspase-3/7 activity between genotypes, the scarcity of apoptotic cells in *sgsh^Δex5−6^* adult sections—and the functional absence of these cells in wild type control tissue—indicates that CNS apoptosis through caspase-dependent mechanisms is not a major feature of the MPS IIIA-like disease state in the zebrafish model. The presence of a clear CNS apoptotic phenotype in early neurodevelopment but not in adulthood may represent an important finding that contextualises the time course of functional neurodegeneration seen in MPS IIIA patients. Perhaps attenuation of early-life features of neuronal degeneration prior to the onset of symptoms may be a viable strategy to mitigate the progression of the disease – however, such an approach would necessitate presymptomatic detection of MPS IIIA disease alleles [92]; this is complicated by the absence of routine newborn screening for MPS III. While the absence of a caspase-dependent apoptotic phenotype in the adult *sgsh^Δex5−6^* zebrafish CNS is in agreement with observations made in an MPS IIIA mouse model [93], this does not preclude the presence of caspase-independent cell death as a driver of MPS IIIA pathology, acting either via lysosomal stress [94] or through factors released by mitochondrial degeneration [95]. Work is ongoing to define involvement of caspase-independent cell death mechanisms in the progression of the MPS IIIA-like state in *sgsh^Δex5−6^* zebrafish. Compared to wild-type CNS axons, which were generally not observed to contain organelles other than abundant mitochondria (Figure 3E–G), the abundance of lysosomes/autophagosomes and generalised intra-axonal disorganisation within *sgsh^Δex5−6^* CNS axons indicates a degree of functional impairment of neuronal activity. These features may be comparable to dystrophic axonal lesions with endolysosomal involvement previously described in MPS IIIA mice [93] and could mechanistically account for the progressive degeneration of neuronal function seen in all MPS III subtypes. Future work will leverage the amenity of the *sgsh^Δex5−6^* zebrafish to live imaging and molecular manipulation to closely examine the cellular and molecular mechanisms driving neuronal dysfunction in MPS IIIA.

Given the detection in the CNS of adult *sgsh^Δex5−6^* zebrafish of large numbers of activated microglial cells, a neuroinflammatory phenotype was suspected. Brain-wide examination of microglial morphology and distribution in larval and adult *sgsh^Δex5−6^* zebrafish demonstrated significant, chronic neuroinflammation persisting from early neurodevelopment into the adult CNS, where the phenotype was exacerbated. Neuroinflammation via stimulation of interleukin-1 has been demonstrated in MPS IIIA mice, where activation of the Toll-like receptor TLR4 by GAGs derived from MPS IIIA mice, but not wild type siblings, drives massive increases in levels of intracellular Il-1β in wild type mixed-glial cultures in a manner similar to induction by LPS stimulation [62]. Contextualising these existing data against the *sgsh^Δex5−6^* model presents two notable findings warranting further investigation. First is the much higher tolerance for chronic LPS-induced inflammation in teleosts such as zebrafish compared to mammals [96]; this is likely an evolutionary adaptation that facilitates the close contact of aquatic species with large numbers of water-borne, potentially pathogenic microorganisms. Second, as a result of teleost-specific genome duplication, zebrafish possess two cognate genes comparable to mammalian TLR4, namely *tlr4ba* and *tlr4bb* (another gene, *tlr4al*, demonstrates high protein similarity to the other two genes but little has been described relating to its function in the literature). However, *tlr4ba* and *tlr4bb* appear to be paralogous rather than orthologous to mammalian TLR4 [97] and do not recognise LPS [98]. As with LPS, HS is known to bind to mammalian TLR4 [99] and induce intracellular production of pro-Il-1β via NF-κB signalling; however, it is presently unclear as to whether or not (i) zebrafish *tlr4* can bind HS despite their inability to bind LPS, or (ii) the neuroinflammation observed in *sgsh^Δex5−6^* zebrafish functions through TLR recognition of HS as a ligand, but primarily through a different—potentially teleost-specific—receptor. In either case, a greater tolerance to chronic neuroinflammation in *sgsh^Δex5−6^* zebrafish relative to mouse MPS IIIA models may explain why homozygous zebrafish mutants are viable well into adulthood despite the onset of diverse neuropathological features. Future experiments will seek to determine the molecular mechanisms of neuroinflammation in the *sgsh^Δex5−6^* zebrafish and how they functionally compare to the recognised TLR4-associated neuroinflammatory phenotype in mammalian MPS IIIA models. To further profile the neuroinflammatory phenotype in *sgsh^Δex5−6^* zebrafish in a manner that was not confounded by divergence in TLR function between teleost and mammalian systems, we instead pharmacologically targeted Caspase-1-mediated activation of the pro-Il-1β zymogen produced following putative HS/*tlr* binding. In order to facilitate cleavage and maturation of pro-Il-1β, Caspase-1 autoactivates within the NLRP3 inflammasome, where it complexes with lysosome-derived Cathepsin B. With continuous treatment of Ac-YVAD-cmk (a potent and irreversible inhibitor of Caspase-1) in larval development, we observed that CNS-wide microglial activation in *sgsh^Δex5−6^* homozygotes was broadly corrected. These data are in accord with elevated Caspase-1 activity detected in MPS IIIA mice [62] and confirm a conserved role for Il-1β-driven inflammation in MPS IIIA neuropathology across evolutionarily distant animal models of the disease. Importantly, however, pharmacological blockade of Il-1β-driven inflammation in *sgsh^Δex5−6^* larvae was only able to rescue a subset of behavioural phenotypes. While basal activity in *sgsh^Δex5−6^* larvae chronically treated with YVAD was closer to that observed in wild-type larvae, this phenotypic rescue was not extended to adverse stimulus-elicited predator-response behaviours. This indicates that the causative elements of the functional neurodegeneration in MPS IIIA are multifactorial, and interventional amelioration of neuroinflammation may be of only partial clinical benefit to affected patients. This is particularly relevant given the ongoing investigation of therapeutic benefits in MPS III patients of pharmaceutical attenuation of neuroinflammation (e.g., an open-label phase-2/3 clinical trial of the recombinant interleukin-1 receptor antagonist Anakinra in MPS III patients, NCT04018755). To this end, future therapeutic options are likely to require simultaneous targeting of diverse aspects of the MPS IIIA disease state in order to elicit a clinically beneficial outcome.

Together, these data signify the presence of a complex CNS disease state in the *sgsh^Δex5−6^* zebrafish that reflects human pathology, validating the *sgsh^Δex5−6^* allele as an excellent experimental model of MPS IIIA with great amenity to a wide variety of future genetic and pharmacological interventions towards both better understanding of MPS III pathobiology and development of much-needed therapeutic avenues.

## 4. Materials and Methods

### 4.1. Animals

Zebrafish were housed and bred in the AquaCore facility, Monash University, according to standard procedures [100]. Larvae were maintained in E3 media (5 mM NaCl, 0.17 mM KCl, 0.33 mM CaCl_2_, 0.33 mM MgSO_4_, supplemented with 1 × 10 − 5% methylene blue as antimycotic), and adults were maintained in system water. Transgenic lines used were Tg(-3.5*ubb*:SECHsa.ANXA5-mVenus)^mq8^ [38]. The *sgsh^Δex5−6^* mutant was generated in our lab in the *Tübingen* wild type background as described below. All experiments were approved by and conducted under the oversight of the Monash University Animal Ethics Committee under ethics approvals ERM14481 and ERM17993.

### 4.2. Generation of sgsh^Δex5−6^ Zebrafish

Two synthetic Alt-R CRISPR RNA (crRNA) sequences targeting exons 5 and 6 of zebrafish *sgsh* were designed using IDT’s Alt-R Custom Cas9 crRNA Design Tool (idtdna.com/site/order/designtool/index/CRISPR_CUSTOM, accessed 19 February 2018). crRNAs were chemically synthesised by IDT, and crRNA sequences and genotyping primers are described in Table 1. All RNAs obtained from IDT were reconstituted to 1 µg/µL in nuclease-free IDT Duplex Buffer (30 mM HEPES, pH 7.5; 100 mM potassium acetate). To generate gRNAs, 5 µL 1 µg/µL of each crRNA was separately annealed to 10 µL of 1 µg/µL IDT Alt-R tracrRNA in a thermocycler; annealing was conducted at 95 °C for 5 min, then ramped down to 25 °C at a rate of 5 °C/min. 1 µL of each annealed gRNA was separately incubated for 10 min at RT with 0.5 µL 10 mg/mL Alt-R *S. pyogenes* Cas9 Nuclease V3 to form functional RNP complexes. A 10 µL dual-gRNA injection solution was then generated, comprising 1.5 µL of each RNP complex, 1 µL Phenol Red as an injection guide, and the remaining volume (6 µL) of the injection solution made up with IDT Duplex Buffer. 0.5–1 nL of the injection solution was injected into zebrafish embryos at the single-cell stage, and injected embryos were raised to adulthood and outcrossed to wild-type zebrafish.

Progeny were screened for germline transmission of the dual-gRNA mediated deletion of *sgsh* exons 5 and 6 by PCR using genotyping primers described in Table 1. PCR parameters for genotyping were as follows: primer annealing at 58 °C, extension time 60 s using GoTaq G2 Green master mix (M7822, Promega, Madison, WI, USA), with primer concentration 500 nM and 30 cycles performed. Sanger sequencing of PCR amplicons from heterozygous F1 progeny with F and R genotyping primers was performed to confirm the presence of the same mutation in all F1 progeny. Subsequently, F1 heterozygotes were in-crossed and homozygous-null F2s were again in-crossed to produce maternal-zygotic null F3 mutants. F3 mutants were used in subsequent experiments.

### 4.3. Reverse Transcription Polymerase Chain Reaction (RT-PCR) of Wild Type and Mutant sgsh Transcript

RT-PCR was conducted using a forward primer targeting *sgsh* exon 1 and a reverse primer targeting *sgsh* exon 7 to confirm the presence of a stable but truncated mRNA product corresponding to the deletion of *sgsh* exons 5 and 6 and the splicing of flanking exons. RT-PCR primer sequences are described in Table 1. cDNA was generated by first-strand synthesis using an equimolar primer mix of Oligo-dT_(18)_ and random hexamers and SuperScript IV reverse transcriptase. Reverse transcription was performed on total RNA extracted from *n* = 50 7 dpf wild type or *sgsh^Δex5−6^* mutant larvae via the TRIzol method. PCR parameters were 30 cycles, with primer annealing at 64 °C using Phusion polymerase in HF buffer with extension time 40 s. Zebrafish *eef1a1l1* was used as a housekeeping control. RT-PCR products were separated using gel electrophoresis in a 1% agarose gel in Tris-acetate EDTA buffer.

RT-PCR amplicons derived from wild type and mutant cDNA were extracted and Sanger sequenced using primers targeting *sgsh* exon 1 or 7, and sequences were aligned against the annotated Ensembl sgsh-201 cDNA sequence (GRCz11 ENSDART00000063147.5) to verify the functional deletion of *sgsh* exons 5 and 6, and the splicing of *sgsh* exons 4 and 7.

### 4.4. In-Silico Modelling of sgsh^Δex5−6^ Monomeric Protein Structure

Using the translate function in the ApE sequence editor program, the amino acid sequence encoded by *sgsh^Δex5−6^* was identified from Sanger sequencing of the mutant RT-PCR product. In silico modelling of both wild type and mutant amino acid sequences was performed using SWISS-MODEL [26] to assess the likely consequences of the deletion. Modelling for both wild type and *sgsh^Δex5−6^* proteins was conducted using the crystal structure of human SGSH (PDB ID 4MHX) [27] as a template.

### 4.5. Sgsh Enzymatic Activity Assay

The enzymatic activity of *Sgsh* in *sgsh^Δex5−6^* mutants compared to wild type controls was determined using a fluorometric assay as previously described [28,29] with minor modifications employed for use with zebrafish tissue. Total protein was extracted from *n* = 100 embryos or whole adult brains or livers from homozygous and heterozygous *sgsh^Δex5−6^* and wild type zebrafish. Tissues were thoroughly homogenised by repeated trituration through a 1000 µL pipette tip followed by a 24 G needle in 1 mL homogenisation solution (0.02 M Tris, 0.5 M NaCl pH 7.4), then sonicated with a SONOPULS mini20 (Bandelin, Berlin, Germany) 2× 30 s on ice. Homogenate was clarified by centrifugation at 11,000× *g* for 5 min, and supernatant was transferred to a new tube. The volume of homogenate was made up to 5 mL with 0.2 M sodium acetate pH 6.5, and concentrated at 4000× *g*, 25 °C using a 10K MWCO centrifugal protein concentrator (88527, ThermoFisher, Melbourne, Australia) until the retentate volume was <1 mL. Buffer exchange was performed by increasing retentate volume to 5 mL with 0.2 M sodium acetate pH 6.5 and re-concentrating to retentate volume <1 mL. This process was repeated three times. Total protein concentration was determined using the Qubit Protein Assay Kit (Q33211, ThermoFisher, Melbourne, Australia) and a DeNovix QFX Fluorometer (DeNovix, Wilmington, DE, USA). Total protein was then diluted to a concentration of 3 µg/µL in 0.2 M sodium acetate pH 6.5. To assay Sgsh activity, 30 µg of total protein was mixed with 20 µL 5 mM 4-Methylumbelliferyl 2-deoxy-2-sulfamino-α-*D*-glucopyranoside sodium salt (4MU-αGlcNS, EM06602, Carbosynth, Compton, Berkshire, UK) dissolved in *Sgsh* activity buffer (0.2 M sodium acetate and 6 µL/mL glacial acetic acid dissolved in distilled water, pH adjusted to 5.8 using 1 M NaOH). The total protein/substrate mix was incubated in 1.5 mL screw-cap tubes at 47 °C for 17 h. This reaction causes desulfation by *Sgsh* in total protein of 4MU-αGlcNS to 4MU-αGlcNH_2_. The first reaction was stopped by the addition of 6 µL 2x McIlvaine’s phosphate/citrate buffer, pH 6.7, containing 0.02% (*w/v*) sodium azide. 0.1 U of *Bacillus stearothermophilus* α-glucosidase (G3651, Sigma, St. Louis, MO, USA) in 10 µL distilled water was then added and a second incubation of 24 h at 37 °C was performed to liberate fluorescent 4-methylumbelliferone (4MU) from the desulfated glucosamine subunit. The second reaction was terminated by the addition of 100 µL 0.5 M carbonate-bicarbonate buffer (0.5 M Na_2_CO_3_/NaHCO_3_, pH adjusted to 10.7 with 5 M NaOH). Samples were immediately transferred to a white, flat-bottomed 96-well assay plate (3912, Corning, New York, NY, USA). Fluorescence data was acquired on a CLARIOstar microplate reader (BMG LabTech, Durham, NC, USA) with excitation 355 nm and emission detection at 460 nm, with 40 flashes applied per well. Gain and focal height were automatically determined by scanning the whole plate and adjusting to capture 90% of signal from the most fluorescent well. Fluorescence values obtained were corrected against readings from a blank reaction where total protein was substituted with MilliQ water (Millipore, Saint-Quentin-en-Yvelines, France). All samples, including blanks, were conducted at least in triplicate. Of note, all *sgsh^Δex5−6^* homozygous groups across all enzyme assays registered a ‘negative fluorescence’ value following blank correction (Figure 1E,F, Appendix A). This is explained by absorbance of the 355 nm excitation wavelength by the total protein input into each reaction. The magnitude of this negative value was highly consistent between replicates within each experiment, but highly variable between different tissue types.

### 4.6. Larval and Adult Behavioural Analysis

Basal behavioural analysis was conducted on 5 dpf zebrafish larvae and 6-month old adults using a custom-built behavioural tracking and analysis system. For larval behavioural analysis, 5 dpf larvae were selected as they exhibit significant spontaneous activity, but still possess yolk cells and are thus not subject to potential behavioural effects of a starvation-like state which may manifest in 7 dpf larvae, in which the yolk has been depleted. Larvae were conditioned to individual wells of a laser-cut imaging chip in 500 µL of E3 medium for 1 h at 28.5 °C. Larvae were then exposed to ambient light for 5 min and the plate was placed into the movement-tracking apparatus. Movement was tracked over a 10 min period. Parameters analysed were movement count, movement duration, and movement distance. For adult behavioural analysis, fish were transferred into a 0.7 L spawning tank (Tecniplast, Italy) half-filled with system water. Through all trials, the water level was maintained at the same height within the imaging tank. Fish were allowed to acclimatise for 5 min at 28 ± 0.5 °C in a temperature-controlled sealed filming box under infrared illumination prior to commencement of video recording. Fish were filmed for a 10 min period following acclimatisation.

For the predator-response assay, 7 dpf larvae treated from 3 to 7 dpf with either 50 µM Ac-YVAD-cmk or 0.1% DMSO (see Methods Section 4.8. below) were used in lieu of 5 dpf so as to capture the full YVAD exposure time-course. Larvae were placed in a custom plate containing 18 individual circular chambers (15.6 mm diameter, 3 mm depth), and acclimatised to the imaging chamber for two minutes prior to commencement of the predator-response assay. The predator-response assay was performed as previously described [101], with minor modifications for use with the custom imaging plates described above. Briefly, larvae were illuminated by a white light from below for six minutes. Subsequently, the white light was switched off, with the resulting 4 min period of darkness simulating environmental threat and predator interaction [101].

For both larval and adult analyses, acquisition of behavioural data was performed using a custom-built digital video imaging system. This apparatus consists of an infrared LED (850 nm) side-lit illumination stage and digital camera mounted on a vibration-free photographic column (Polaroid M3, Polaroid Inc.). Infrared imaging was performed using a BlackMagic Micro Studio 4K digital camera (BlackMagic Design, Australia) equipped with an 850 nm filter. The camera was paired with a true 1:1 Macro objective lens with focal length 30 mm (Olympus, Japan). Native videos of 10 min duration were recorded at standard resolution of 1920 × 1080 pixels (1080p) and a framerate of 24 frames per second. All video files were acquired using an external HDMI recorder (Atomos Shogun, Melbourne, Australia) equipped with a programmable time-resolved video acquisition functionality. Native files were saved in .mov digital containers and encoded with a ProRes 422 HQ coded that provided no temporal compression artefacts (interframe-only encoding) and variable bitrate.

For analysis of behavioural data, a standardised method was used [31]. Video files were post-processed using DaVinci Resolve 15 (BlackMagic Design, Australia) video editing software. Post-processing involved sharpening, baseline contrast correction, and background subtraction. Corrected files were then analysed using EthoVision XT v15 tracking software (Noldus Information Technology, Wageningen, the Netherlands, accessed 31 October 2019). Digital video-based tracking of animal behaviour was based on a reconstruction of movement pattern analysis in a grid of pixels on individual frames of the video files. Software algorithms analysed each frame of the video file to distinguish the tracked animals from the background. This was performed based on semi-automated adjustment of threshold of pixel intensity and colour saturation values. Each detected animal was automatically assigned a mathematical centre of gravity (centroid) derived from the average surface area. Automatic frame-by-frame tracking produced time-stamped *x,y* coordinate pairs assigned to centroids of detected objects and provided a foundation for the reconstruction of graphical animal trajectories, averaged occupancy heatmaps and behavioural parameters (i.e., average distance travelled, average speed, number of active organisms) calculated for each test chamber. Zones of interest were defined in order to calculate the edge-preference index of the fish.

### 4.7. Caspase-3/7 Activity Assay

To quantify Caspase-3/7 activity in the adult brain of wild type and *sgsh^Δex5−6^* zebrafish, the Caspase-Glo^®^ 3/7 luminescence assay (G8090, Promega, Madison, WI, USA) was employed. Each genotype was assayed in triplicate, with each replicate comprising *n* = 3 brains. Briefly, cells from freshly-dissected adult zebrafish brains were dissociated into single cells via a 15-min digestion in 1 mg/mL collagenase-II in 1× HBSS (with Ca and Mg) with 50 µM EDTA pH 8.0 at 37 °C, with gentle manual trituration every 5 min through a trimmed P1000 pipette tip. Cells were pelleted and washed three times with cold 1× HBSS + 2% inactivated fetal calf serum (FCS), then quantified on a Countess Automated Cell Counter (Invitrogen, Grand Island, NY, USA) with Trypan Blue staining using standard protocols to determine cell viability. Caspase activity was assayed per manufacturer’s instructions, with 50,000 cells assayed per sample in triplicate at both 30 and 60 min of incubation. In addition to experimental reactions, a blank control reaction (Caspase-Glo^®^ 3/7 reagent with 1× HBSS + 2% inactivated FCS) was performed, and all luminescence values for experimental groups were corrected against the luminescence of the blank reaction. Luminescence was recorded on a BMG CLARIOstar Plus microplate reader (BMG LabTech).

### 4.8. Larval Ac-YVAD-cmk Treatment

Wild type and *sgsh^Δex5−6^* larvae were raised under standard conditions to 3 dpf, then continuously incubated in either E3 media containing 50 µM Ac-YVAD-cmk (Acetyl-tyrosine-valine-alanine-aspartate-chloromethyl ketone, SML0429, Sigma, St. Louis, MO, USA), or E3 with 0.1% DMSO as vehicle control, until 7 dpf. Fresh drug or vehicle control was applied each day of exposure.

### 4.9. LysoTracker Staining of Zebrafish Larvae

Wild type and *sgsh^Δex5−6^* homozygous zebrafish larvae were raised to 7 dpf, then incubated for 40 min in 10 µM LysoTracker Red DND-99 (L7528, ThermoFisher, Rockville, MD, USA) in E3 medium in a 15 mL Falcon tube, at 28.5 °C in darkness. The staining solution was subsequently replaced twice with fresh E3, each for five minutes of washing prior to mounting of larvae in 0.5% low-melting temperature agarose in E3. Stained larvae were live-imaged dorsally using a Zeiss LSM 710 confocal microscope with excitation 555 nm and band-pass detection filters gated between 580 and 610 nm.

### 4.10. Tissue Processing for Section Immunohistochemistry

Following sacrifice by immersion in an ice-water slurry, adult zebrafish were exsanguinated by severing the posterior cardinal vein at the level of the anal pore; the dorsal neurocranium was then carefully resected to expose the brain which was left in situ. Fish were then decapitated at the level of the pectoral fins. Heads were transferred into ice-cold 2% paraformaldehyde (PFA, 158127, Sigma, St. Louis, MO, USA) in phosphate buffer pH 7.4, and incubated overnight at 4 °C with slow shaking. Samples were then transferred to a cryoprotection/decalcification solution (20% sucrose, 20% 0.5 M EDTA in 1× PBS) and incubated overnight at 4 °C with slow shaking, then cryo-embedded in sucrose-gelatin embedding medium as previously described [70]. Serial sections (16 µm thickness) of the whole brain were obtained using a Leica CS3080 cryostat.

### 4.11. Immunohistochemistry

Frozen sections were allowed to dry at room temperature (RT) for >1 h, then rehydrated in 1× PBS for 15 min. Sections were then permeabilised 2× 15 min in 1× PBS with 0.3% Triton X-100 (PBS-Tx), and incubated flat in a humidified chamber with primary antibody overnight at 4 °C. Sections were then gently washed 3× 20 min in PBS-Tx at RT, and incubated with secondary antibody in PBS-Tx with DAPI at 1:5000 concentration for 60 min at RT. One 10 min wash with PBS-Tx was performed, followed by 2× 20 min washes in 1× PBS at RT. Slides were then mounted with 50% glycerol in 1× PBS and coverslipped prior to imaging. Sections were imaged on a Leica TCS SP8 confocal microscope equipped with a HyD detector with bandpass filters set for the necessary fluorophore(s).

### 4.12. Wholemount Immunochemistry

Larvae were euthanised by an acute overdose of Tricaine (ethyl 3-aminobenzoate methanesulfonate, E10521, Sigma Aldrich, St. Louis, MO, USA) and transferred immediately to ice-cold 2% PFA in phosphate buffer. Following overnight incubation at 4 °C with slow shaking, fixed larvae were transferred to 1× PBS for dissection of larval brains. Briefly, dissections consisted of removing both eyes by severing the optic nerve, then peeling the skin from the dorsal surface of the cranium to expose the brain. More caudal aspects of the brain were exposed by carefully removing anterolateral skeletal muscle, and the brain was lifted away from the developing cribriform plate and finally resected by severing at the level of the anterior-most spinal cord. All subsequent incubation steps, either at RT or 4 °C, were conducted with slow shaking. Larval brains were subsequently washed twice in PBS-Tx 0.5% for 30 min at RT, then preincubated in a blocking solution composed of 1% DMSO and 10% normal goat serum in PBS-Tx 0.5% for at least 2 h at RT. Rabbit anti-L-plastin primary antibody (a kind gift from Michael Brand, Centre for Regenerative Therapies, Technische Universität Dresden) was applied overnight at 4 °C at 1:2500 concentration with 2% normal goat serum in PBS-Tx 0.5%. Samples were then washed once for 10 min and 3× for 30 min at RT in PBS-Tx 0.5%, then pan-isotype goat anti-rabbit IgG Alexa Fluor 488 secondary antibody (A27034, ThermoFisher, Waltham, MA, USA) was applied at 1:1000 overnight at 4 °C. Brains were then washed once for 10 min and twice for 60 min in PBS-Tx 0.5%, then once for 10 min in 1× PBS, and subsequently passed through a glycerol series with ascending concentrations to 70% in 1× PBS. Samples were then mounted in 70% glycerol in 1× PBS and confocally imaged on a Leica TCS SP8 with a 20× immersion lens. For both wholemount and section-based detection of microglia by L-plastin immunoreactivity, inactive microglia were identified on the basis of possessing a ramified morphology, while active microglia were identified on the basis of possessing an amoeboid morphology.

### 4.13. Tissue Processing for Transmission Electron Microscopy

Adult zebrafish were sacrificed as described above. Following resection of the neurocranium, fish were transferred to 1× PBS and the optic nerves were severed immediately behind the retinae so as to prevent the optic chiasm tearing the forebrain away from the optic tectum when the brain was dissected out of the cranium. Tissue ventral to the brain was carefully dissected away to expose the whole brain, which was then resected by a single incision through the anterior spinal cord, then sub-dissected into fore-, mid- and hindbrain regions. Spinal cord was separately dissected in a single piece by careful removal of vertebrae. Tissue was immediately transferred into Karnovsky fixative (2.5% glutaraldehyde and 2% PFA, with 0.25% CaCl_2_ and 0.25% MgCl_2_ in 0.1 M sodium cacodylate, pH 7.4), and fixed at RT for 2 h, then post-fixed in 1% osmium tetroxide/1.5% potassium ferricyanide followed by 5× 10 min washes in distilled water. Tissue was then incubated in 70% ethanol overnight, then dehydrated in a stepwise, increasing ethanol gradient (2× 10 min washes for 80, 90 and 95% ethanol, then 4× 10 min washes in 100% ethanol). Samples were embedded in Epon 812 embedding resin. Ultrathin sections (70 nm) were cut using a Leica UltraCut UC7 (Leica, Austria) and stained with uranyl acetate and lead citrate. Sections were imaged at 80 kV on a Jeol 1400+ transmission electron microscope.

### 4.14. Mass Spectrometry

LC-MS/MS quantification of HS was performed as previously described [30] at the Mass Spectrometry Core Facility, South Australian Health and Medical Research Institute (SAHMRI, Adelaide, SA, Australia). Tissue samples were homogenised in 1 mL 10% MeOH using a Precellys 24 Tissue Homogenizer (Bertin Instruments, Paris, France), 2× 20 s at 6500 rpm at 4 °C with a 30 s break between. Total protein concentration was determined using a BCA assay. 50 µg total protein were placed in glass tubes and diluted to the same volume (229 µL) in 10% MeOH. A standard curve was constructed, comprised of disaccharide GlcN α1–6 GlcUA at concentrations 1, 5, 20, 100, 500, 1000 and 1500 ng, with standards also diluted in 10% MeOH and freeze-dried overnight. 50 µL 2,2-dimethoxypropane followed by 1 mL 3M butanolic HCl was added to all tubes, which were then sealed and incubated at 100 °C for 2 h. After incubation, samples were dried under N_2_ for 1 h at 45 °C, then reconstituted in 200 µL deuterated HS internal standard. This was mixed for 30 min on an orbital shaker (Ratek Instruments), then centrifuged for 15 min at 13,000 rpm on a Biofuge pico (Heraeus, Hanau, Germany). The supernatant was transferred to a 96-well microtitre plate and placed at 6 °C in a Waters Acquity UPLC sample manager until analysed. MS was performed using an API 4000 QTrap mass spectrometer (AB/Sciex, Framingham, MA, USA). LC separation prior to MS analysis was performed using an Acquity UPLC (Waters Corporation, Milford, MA, USA) equipped with a 2.1 mm × 50 mm BEH C18 (1.7 µm particle size) analytical column (P/N. 186002350, S/N. 03313900315181). For LC, a binary solvent system was used with solvent A consisting of water with 0.1% formic acid, and solvent B consisting of acetonitrile with 0.1% formic acid. A 2 µL injection of sample was loaded onto the column at a flow rate of 350 µL/min using 80% solvent A. Chromatographic separation was performed using the gradient shown in Table 2.

For MS, data were acquired in MRM (multiple reaction monitoring) mode. In this mode, transitions were monitored at 50 ms scan-time each, *m/z* 468.245 to 162.077 (GlcN-GlcUA, butylated HS), *m/z* 477.300 to 162.077 (d_9_ disaccharide IS). The peak areas were calculated using Analyst 1.6.2 (AB/Sciex, Framingham, MA, USA). All samples were analysed in a random order interspersed every three injections with a blank injection of milli-Q water. There are multiple peaks observed with butanolysis of HS. The dominant peak relating to the GlcN α1–6 GlcUA disaccharide was integrated and used as a marker for total HS.

### 4.15. Statistical Analyses

Statistical analyses for all experiments were conducted in GraphPad Prism 9 (GraphPad Software). All data were first analysed for normality with Anderson-Darling, D’Agostino, Shapiro-Wilk and Kolmogorov-Smirnov tests. Unpaired T-tests or Mann-Whitney U tests were used to determine significance for experiments that compared two groups, with normality testing performed for all data to determine whether or not data was parametric. Ordinary one-way ANOVA with Tukey’s post-hoc multiple comparisons test or Brown-Forsythe and Welch ANOVA with Dunnett’s T3 multiple comparisons test was used to determine statistical significance in assays comparing more than two groups. In all experiments, data were considered significant if *p* < 0.05.

## Figures and Tables

**Figure 1 ijms-22-05948-f001:**
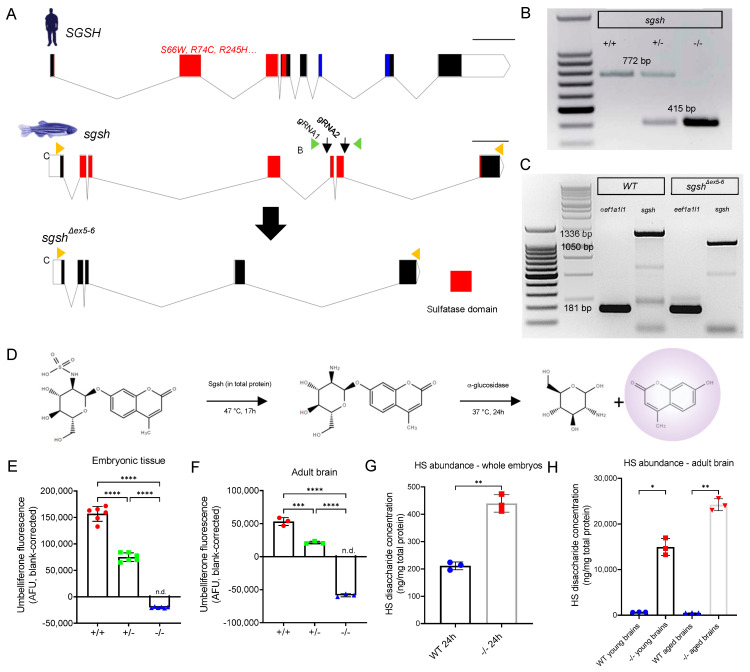
Generation and biochemical validation of the *sgsh^Δex5−6^* mutant zebrafish. (**A**) Exon-intron diagram representations of wild type human *SGSH*, wild type zebrafish *sgsh*, and the zebrafish *sgsh^Δex5−6^* allele. Exons comprising sulfatase domain highlighted in red; exons comprising sulfatase-associated Domain of Unknown Function DUF4976 highlighted in blue. Downwards-facing arrows indicate gRNA target sites. Orange arrowheads reflect primer positions for RT-PCR; green arrowheads reflect primer positions for genotyping PCR. Scale bar is 1 kilobase. (**B**) Genotyping PCR of wild type (+/+), heterozygous (+/−) and homozygous *sgsh^Δex5−6^* (−/−) alleles. Wild type *sgsh* produces a band at 772 bp, while *sgsh^Δex5−6^* produces a band at 415 bp. (**C**) RT-PCR amplicons of *sgsh* in RNA from wild type (+/+) or homozygous *sgsh^Δex5−6^* (−/−) zebrafish embryos at 24 hpf using primers located in the first and last exons of *sgsh*. Wild type *sgsh* produces a prominent band at 1336 bp, while *sgsh^Δex5−6^* produces a band at 1050 bp corresponding to loss of exons 5 and 6. *eef1a1l1* (amplicon size 181 bp) included for both genotypes as housekeeping control. (**D**) Overview of Sgsh enzyme activity assay. 4MU-αGlcNS is desulfated by Sgsh in total protein, then glucosamine is hydrolysed from UV-fluorescent 4MU by bacterial α-glucosidase. (**E**) Enzyme activity in wild type (+/+), *sgsh^Δex5−6^* heterozygous (+/−) and homozygous (−/−) whole embryos (*n* = 100 embryos per sample, 6 samples per group). (**F**) Enzyme activity in adult brain (*n* = 3 brains per sample, 3 samples per group) for each genotype. Negative arbitrary fluorescence values observed in *sgsh^Δex5−6^* homozygous samples result from absorbance of UV excitation wavelengths by protein input and blank correction. Data in (**E**,**F**) are presented as mean ± SEM and tested via ordinary one-way ANOVA with Tukey’s multiple comparisons test; **** *p* < 0.0001 and *** *p* = 0.0001. (**G**) HS disaccharide abundance in whole 24 hpf embryos from either wild type or *sgsh^Δex5−6^* genotypes as determined by LC-MS/MS (*n* = 50 embryos per sample, *n* = 3 samples per group). Data presented as mean ± SEM and tested by Welch’s *t*-test. ** *p* = 0.0021. (**H**) HS abundance in young (6-month-old) or aged (18-month-old) brains from either wild type or *sgsh^Δex5−6^* genotypes determined by LC-MS/MS (*n* = 3 brains per sample, *n* = 3 samples per group). Data presented as mean ± SEM and tested by Brown-Forsythe and Welch ANOVA with Dunnett’s T3 multiple comparisons test. * *p* = 0.0171, ** *p* = 0.0031.

**Figure 2 ijms-22-05948-f002:**
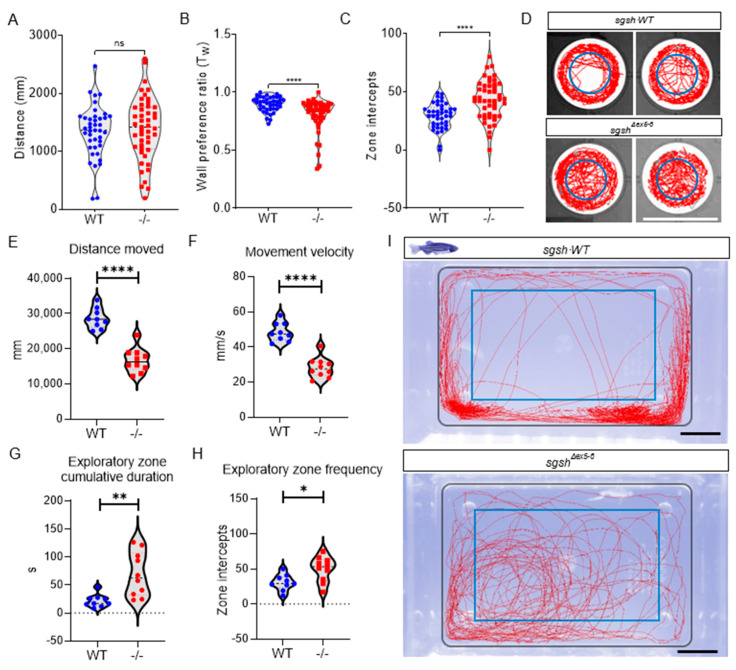
*sgsh^Δex5−6^* zebrafish exhibit progressive behavioural abnormalities. Behavioural tracking of 5 dpf wild type and *sgsh^Δex5−6^* homozygous larvae, examining (**A**) cumulative distance travelled, (**B**) wall preference ratio (T_w_), and (**C**) zone intercepts to and from the central arena reflecting exploration behaviour. In (**A**–**C**) each point represents an individual larva. (**D**) Representative movement traces of single larval movement in the imaging arena throughout the recording period for both wild type and *sgsh^Δex5−6^* genotypes. Central blue circle indicates region for zone intercept calculations. Scale bar 15 mm. (**E**) Cumulative distance travelled by adult (6-month-old) wild type and *sgsh^Δex5−6^* zebrafish. (**F**) Average movement velocity between adult wild type and *sgsh^Δex5−6^* zebrafish. (**G**) Cumulative time spent in exploratory zone between adult wild type and *sgsh^Δex5−6^* zebrafish. (**H**) Frequency of interception with exploratory zone boundaries between adult wild type and *sgsh^Δex5−6^* zebrafish. Data in (**A**–**C**) are presented as mean ± SEM and tested by Mann-Whitney U test, and in (**E**–**H**) by unpaired *t*-test. In all samples **** *p* < 0.0001 and n.s. *p* > 0.05; in G ** *p* = 0.0029 and in H * *p* = 0.0146. (**I**) Representative movement traces of single adult movement in the imaging arena throughout the recording period for both wild type and *sgsh^Δex5−6^* genotypes. Inner blue rectangle indicates region for exploratory zone calculations. Scale bars 1 cm.

**Figure 3 ijms-22-05948-f003:**
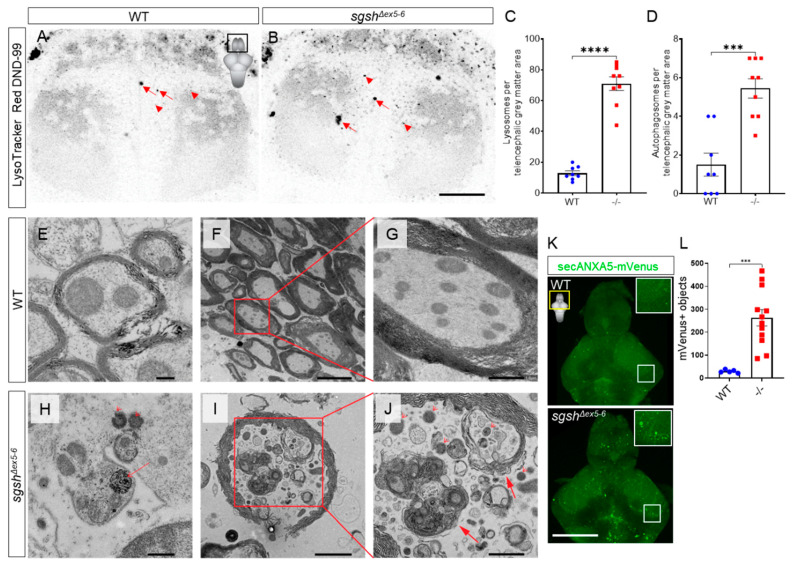
Cellular pathological features of MPS IIIA are recapitulated in *sgsh^Δex5−6^* zebrafish. Single-plane images of LysoTracker Red DND-99 live staining of lysosomes (arrowheads) and autophagosomes (arrows) in the telencephalon of larval zebrafish of either wild type (**A**) or *sgsh^Δex5−6^* (**B**) genotypes at 7 dpf. Scale bar = 20 µm. (**C**) Quantification of LysoTracker^+^ lysosomes in dorsal telencephalic grey matter between wild type and *sgsh^Δex5−6^* larvae. (**D**) Quantification of LysoTracker^+^ autophagosomes in dorsal telencephalic grey matter between wild type and *sgsh^Δex5−6^* larvae. Each point represents a single telencephalon. Data in (**C**–**D**) presented as mean ± SEM and tested by unpaired *t*-test; *** *p* = 0.0001 and **** *p* < 0.0001. (**E**–**G**) Ultrastructural examination of cerebellar axons in aged (18-month-old) wild type brain. Scale bars 200 nm in (**E**), 5 µm in (**F**) and 1 µm in (**G**). Intracellular (**H**) and axonal (**I**–**J**) lysosome (arrowheads) and autophagosome (arrows) overabundance in cerebellar neurons from aged *sgsh^Δex5−6^* brain. Scale bars 500 nm in (**H**), 2 µm in (**I**) and 1 µm in (**J**). (**K**) Detection of apoptotic objects as labelled by the Tg(-3.5*ubb*:SECHsa.ANXA5-mVenus) transgene in 7 dpf wild type and *sgsh^Δex5−6^* larval brains; scale bar 50 µm. Insets are boxed region in each panel. (**L**) Quantification of ANXA5-mVenus apoptotic objects in 7 dpf wild type and *sgsh^Δex5−6^* larval brains. Each point represents a single brain. Data presented as mean ± SEM and tested by Mann-Whitney U test; *** *p* = 0.0003.

**Figure 4 ijms-22-05948-f004:**
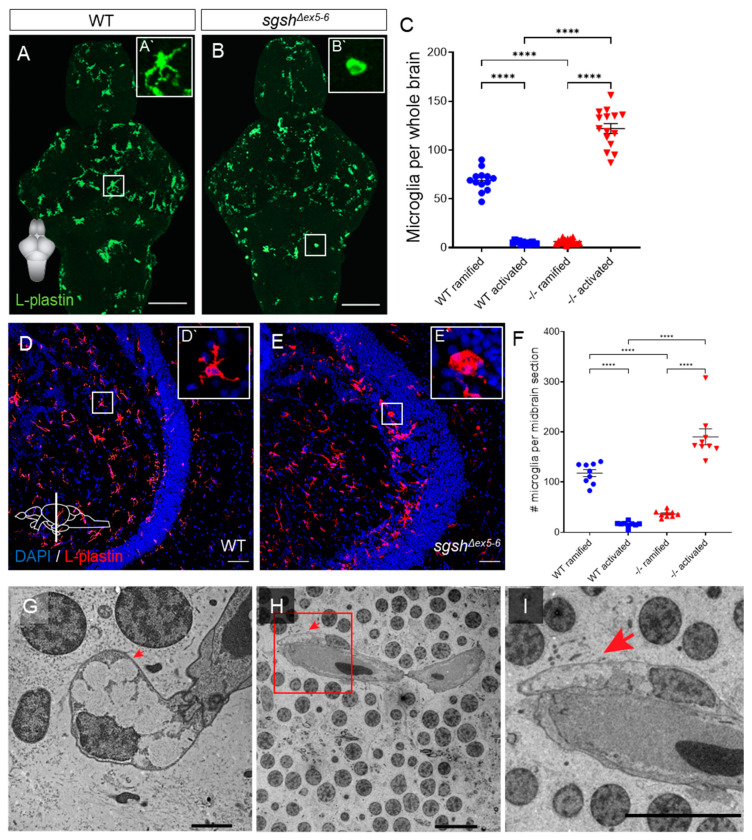
Microglial activation and neuroinflammation in the larval and adult *sgsh^Δex5−6^* CNS. (**A**,**B**) L-plastin immunostaining of microglia in wild type and *sgsh^Δex5−6^* 7 dpf larval brains. Scale bar = 50 µm. (**A`**) Zoom of boxed region in (**A**) showing a representative ramified microglial cell in the larval zebrafish brain. (**B`**) Zoom of boxed region in (**B**) showing representative amoeboid microglial cell in the larval zebrafish brain. (**C**) Quantification of L-plastin^+^ microglial morphology and abundance in wild type or *sgsh^Δex5−6^* 7 dpf brains. Each point represents a single brain. Data presented as mean ± SEM and tested by ordinary one-way ANOVA with Tukey’s multiple comparisons test; **** *p* < 0.0001. (**D**–**E**) L-plastin immunostaining of microglia in midbrain sections of adult wild type and *sgsh^Δex5−6^* zebrafish, scale bar = 50 µm. (**D`**) Zoom of boxed region in (**D**) showing representative ramified microglial cell in the adult zebrafish midbrain. (**E`**) Zoom of boxed region in (**E**) showing representative amoeboid microglial cell in the adult zebrafish midbrain. (**F**) Quantification of L-plastin^+^ microglial morphology and abundance in adult wild type and *sgsh^Δex5−6^* midbrain sections. Each point represents a single section, *n* = 3 sections of approximately equal level quantified per brain. Data presented as mean ± SEM and tested by ordinary one-way ANOVA with Tukey’s multiple comparisons test; **** *p* < 0.0001. (**G**–**I**) Detection by TEM of perivascular microglia with activated morphology with large, electron-lucent intracellular vacuoles (arrows) in the midbrain of 18-month-old *sgsh^Δex5−6^* zebrafish. Scale bars 2 µm in (**G**), 100 µm in (**H**,**I**).

**Figure 5 ijms-22-05948-f005:**
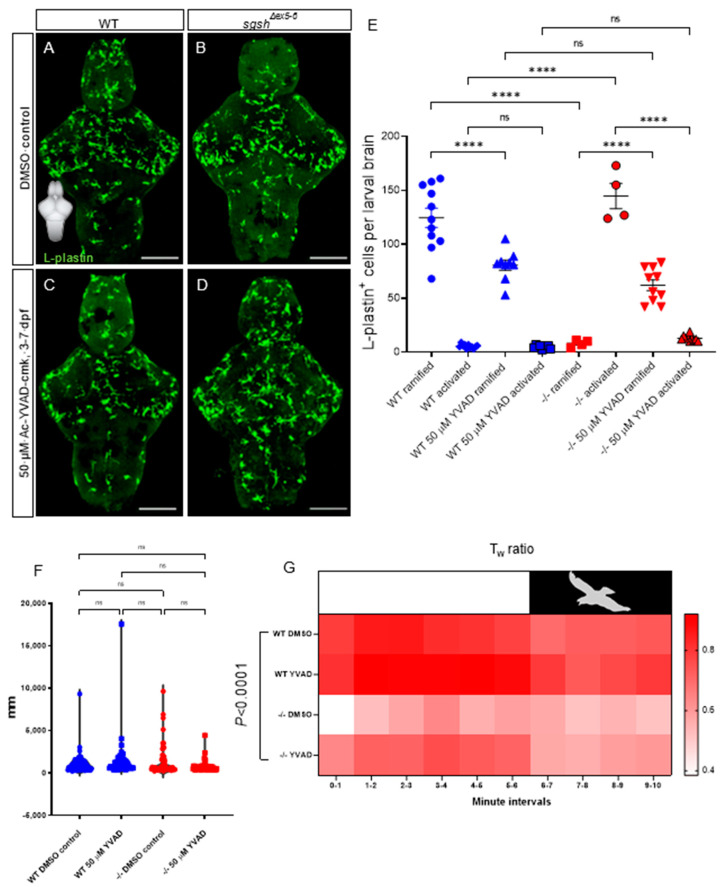
CNS neuroinflammation in *sgsh^Δex5−6^* larvae is corrected by continuous pharmacological Caspase-1 blockade. (**A**,**B**) L-plastin immunostaining of microglia in 7 dpf wild type and *sgsh^Δex5−6^* brains from larvae continuously treated from 3–7 dpf with 0.1% DMSO as control. (**C**,**D**) L-plastin immunostaining of microglia in 7 dpf wild type and *sgsh^Δex5−6^* brains from larvae continuously treated from 3–7 dpf with 50 µM Ac-YVAD-cmk Caspase-1 inhibitor. Scale bars 50 µm. (**E**) Quantification of L-plastin^+^ microglial morphology and abundance following treatment with either 50 µM Ac-YVAD-cmk or 0.1% DMSO in wild type and *sgsh^Δex5−6^* larval brains at 7 dpf. Each point represents a single brain. Data presented as mean ± SEM and tested by ordinary one-way ANOVA with Tukey’s multiple comparisons test; **** *p* < 0.0001, n.s. *p* > 0.05. (**F**) Neither YVAD nor DMSO exposure significantly alters basal movement (cumulative distance travelled, mm, in constant illumination) in either wild type or *sgsh^Δex5−6^* 7 dpf larvae. Data presented as mean ± SEM and tested by one-way ANOVA with Tukey’s multiple comparisons test; n.s. *p* > 0.05. (**G**) Chronic YVAD administration to *sgsh^Δex5−6^* partially rescues impaired thigmotaxis behaviours under standard conditions (white boxed region, representing constant levels of light), but does not attenuate the impaired escape response in response to sudden threat conditions (black boxed region, representing sudden switching off of light and resulting darkness). Data presented as heat map reflecting relative mean T_w_ (wall preference ratio) metric at specific time point of sample recording. Data were analysed by two-way ANOVA with stacked matching, where sphericity of variances was not assumed and α = 0.05. The treatment condition was found to represent 47.43% of variation in the dataset, *p* < 0.0001. Each group represents mean values of 18 replicate trials per each condition.

**Table 1 ijms-22-05948-t001:** Primer and crRNA sequences used in generation of the *sgsh^Δex5−6^* mutant.

Name (IDT Identifier)	Sequence (PAM Not Included for crRNAs)
*sgsh* crRNA 1 (CD.Cas9.JJFS2685.AM)	5′-TCCGGACACTCCTGCAGCGA-3′
*sgsh* crRNA 2 (CD.Cas9.JJFS2685.AB)	5′-CCAGGCCTACGTCAGTCTAC-3′
*sgsh* genotyping F primer	5′-GGATGTCTTTATCCCGAACATAAC-3′
*sgsh* genotyping R primer	5′-ATTTACTCACCCTTGGACCATTTA-3′
*sgsh* RT-PCR exon 1 F primer	5′-TGATGTAGGAGGATGCAGAAGTAG-3′
*sgsh* RT-PCR exon 7 R primer	5′-AGCCAGATTCACCTTCTCCA-3′
*eef1a1l1* RT-PCR F primer	5′-GAAGACAACCCCAAGGCTCTC-3′
*eef1a1l1* RT-PCR R primer	5′-CCACCGATTTTCTTCTCAACG-3′

**Table 2 ijms-22-05948-t002:** Chromatographic gradient used for the analysis of butanolic products of HS.

Time	% A	% B	Flow (mL/min)
0.00	80	20	0.350
3.50	60	40	0.350
3.51	1	99	0.350
4.50	80	20	0.350
5.50	80	20	0.350

## Data Availability

The datasets generated and/or analysed in the present study are available from the corresponding author upon reasonable request.

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
