# Peer review of "An Engineered sgsh Mutant Zebrafish Recapitulates Molecular and Behavioural Pathobiology of Sanfilippo Syndrome A/MPS IIIA"

_ijms, 2021, doi:10.3390/ijms22115948_

Round 1
Reviewer 1 Report
Douek and co-workers present a report on the development of a zebrafish model for Sanfilippo syndrome A (MPS IIIA), which nicely recapitulates the biochemical and behavioural pathology that characterizes the human disease.
This is an extremely elegant study, which relied in a state of the art technology for targeted mutagenesis and model generation, while taking into account recent data on transcriptional adaptive compensatory responses, to generate a transcriptionally stable but completely non-functional Sgsh gene product. The model was extensively characterized and nicely shown to recapitulate a series of CNS-specific features of MPS IIIA including complex behavioural phenotypes and profound, lifelong neuroinflammation.
The manuscript is well written, methods were carefully designed and the results and discussion clearly presented.
I believe this will be a model of upmost value to further characterize MPS IIIA pathophysiology and a great tool for drug discovery. I have no doubts in recommending it for publication just the way it is.
Author Response
Reviewer 1 recommended publication as-is and thus did not have any comments requiring revision. We thank Reviewer 1 for their time taken to read our manuscript.
Reviewer 2 Report
The authors present a novel zebrafish knockout mutant model of Sanfilippo A in zebrafish using CRISPR-Cas9 technology which aims to replicate the human characteristics of disease. They report key changes in biochemical primary storage which increases with age and changes in behaviour activity and neuroinflammation. This model is going to be useful to understand genotype/phenotype correlations and useful for high throughout analysis of behaviour and potential treatments. Whilst there is extensive analysis conducted by the authors there are some key flaws and additional experiments needed in the paper before I can commend this paper for publication.
- Gangliosides and secondary lipid changes have been previously shown in a number of studies to be contributory to the neurocognitive decline MPS IIIA disease pathology . Whilst the authors characterised the primary HS storage, this study could benefit from additional characterise of GM2 and GM3 gangliosides in particular and the timing of these in the zebrafish model.
- The 7 dpf timepoint presented throughout the manuscript seems to be an early embryonic timepoint not really reflective of disease pathology with zebrafish lifespan being 3-5 years. The analysis of lysosomes and autophagosomes is likely to be observed as the disease progresses and later timepoints should be analysed throughout. Please justify the use of the 7 dpf timepoint and how this translates to mouse and human MPS IIIA disease. We know that there are clear differences between mouse and human models developmentally and this seems to be a very early timepoint to assess disease pathology.
- The exon structure of the SGSH gene appears to be different in the zebrafish and human as shown in Figure 1A. Most patient mutations are point mutations and small indels rather than truncated proteins as replicated in this model. Comment on translatability to humans.
- The authors report the measurement of SGSH using the 4MU fluorogenic method. Did they repeat this using the radiolabelled substrate method to determine if the background issues was due to the fluorogenic nature of the substrate.
- Were other lysosomal enzymes measured to determine the specificity of the enzyme activity knockdown. Other enzymes such as beta-hexosaminidase have been previously been shown to be secondarily elevated in mouse models of MPS IIIA. These should be reported in addition to SGSH.
- The authors report changes in activity in the zebrafish in the larvae (Fig 2D) and a decrease in activity at 6m of age (Fig 2E and 2F). Did they measure the activity at aged zebrafish at 18 months when they also did biochemistry. How do these correlate to mouse and human MPS IIIA disease?
- Were zebra bodies characteristic in lysosomal storage EM sections observed in Figure 3?
- Figure 3A and B is very difficult to see the difference in morphology between the lysosomes and autophagosomes. Can this figure be in colour as it’s Red dye tracking system? Were other areas of the brains analysed or just the telecephalon?
- The authors report throughout the manuscript changes in microglia activation. Was astrocyte activation also examined? This has been shown to be a key part of the neuroinflammatory cascade in MPS IIIA and many papers have analysed markers of astrocytosis (GFAP) and microglia (isolectin B4). How does L-plastin compare to isolectin B4 and why was this chosen as a microglia marker? Was L-plastin staining conducted at 18mths in older animals and quantitated?
- The Ilb1 blockade using YVAD is interesting. However analysis was only conducted using early life exposure and analysis at 7dpf. In order to determine if this has an effect on the neurological disease pathology at least adult (6 mths) and aged (18 mths) when neuroinflammation is present should be analysed. The chronic neuroinflammation alluded to by in the authors and in section 2.5 was only analysed at 7 dpf and longer timepoints need to be included.
- Why was the behaviour testing in section 2.5 done differently to the exploratory data presented in Fig 2? This has only been reported again in the 7 dpf zebrafish where little differences, except centre crossing was observed. This manuscript would also benefit from additional learning and cognitive testing to determine if the zebrafish do recapitulate the neurocognitive decline seen in both mice and human disease in addition to changes in activity.
- Was there any mortality or growth defects associated with the YVAD treatment in either wildtype or MPS IIIA?
- Was LAMP antibody staining done to compare to Lysotracker?
- Is there any effect on end stage pathology survival in the zebrafish. How long do they survive for compared to wildtype?
- Anakinra an Il1b antagonist is in clinical trial for MPS IIIA this should be mentioned in the introduction and discussion.
Minor revisions
Page 2, line 77-78. Mention of MPS IIID and MPS IIIE should be made including references for completeness
Figure 2B. Tw should be written in full on the Y-axis.
Figure 2D and 2I. Scale bars should be included on the behaviour swimming task maps.
Figure 3K. A higher magnification of this figure should be included. Comment on the use of young zebrafish to analyse this? Apoptosis whilst a dynamic process in development is likely to be altered in older SGSH zebrafish compared to younger ones.
Figure 4G. Has this been shown to be HS storage? Or just assumed? It looks like characteristic foamy vacuoles seen in lysosomal storage. The legend should reflect this.
Figure 5G. X-axis should be units of minutes.
Discussion. Line 477-479. Additional references for behaviour testing need to be included to be reflective of the literature.
Author Response
Please find our reviewer comments attached. We thank Reviewer 2 for their in-depth, useful and helpful comments, which we have broadly implemented and will inform future work.

Reviewer 3 Report
Review
The manuscript “An engineered sgsh mutant zebrafish recapitulates molecular 2 and behavioral pathobiology of Sanfilippo syndrome A/MPS 3 IIIA” by Douek et al. described a generation of the first zebra fish gene targeted model of Sanfilippo type a syndrome (MPS IIIA). Using CRISPR-Cas9 mutagenesis authors generated a strain expressing SGSH protein mutant in MPS IIIA lacking exons 5 and 6. The homozygous animals show a complete deficiency of the enzymatic SGSH activity in tissues and store high levels of heparan sulfate. Importantly adult fish and larvae show clear behavioral abnormalities resembling those of human patients and mouse model (hyperactive behavior followed by hypoactive one, sign of reduced anxiety) and a pronounced levels of neuroinflammation. Moreover the authors demonstrate that the levels of neuroinflammation can be reduced by pharmacological inhibition of Caspase1 involved in generation of interleukin-1β which partially rescues behavioral deficits.
Overall this is an interesting study describing a valid novel animal model highly suitable for high-throughput drug screening. Still the manuscript needs to be improved before publication as described below.
Major changes
Figure 1 E and F. The graphs showing SGSH activity in fish tissues are somewhat awkward. First the activity should be expressed not as obituary fluorescence units, but rather as the units of enzymatic activity (nmol of product liberated per hour (or per minute) per mg of protein in the tissue extract). Besides, the blank samples used by authors (“a blank reaction where total protein was substituted with MilliQ water”) are not appropriate and need to be supplemented with protein extracts added after the reactions are terminated by carbonate-bicarbonate buffer. This would allow to avoid obtaining negative values in the samples from the gene-targeted fish.
Supplementary Figure 3C and 3D. Personally, I am not convinced that the micrograph presented by the authors shows apoptotic bodies. They should label the panel and explain which features they used to recognize the apoptotic bodies and activated microglia cells on the EM microphotographs.
Figure 4. For larvae, the WT brain is shown on the left panel and KO on the right, while for the adult fish its’ vise-versa, besides panels are not labeled, which adds to the confusion. Importantly, some microglia on the right panel (WT?) clearly show amoeboid morphology. Can the authors use antibodies that would mark only activated microglial cells? Would fish have analogues to CD11b or Iba1a? Would activated microglia be stained with ILB4, like activated mouse or human microglial cells?
Minor changes.
Line 78 and below: “MPS IIIA, the most common of the MPS III subtypes [16], tends to exhibit greater clinical severity than other MPS III subtypes, possibly due to its upstream positioning in the HS degradation pathway and the lack of endogenous compensatory activity by other enzymes.” This conclusion is rather speculative and needs to be supported by references.
Figure 1D. The fluorescent assay described by the authors is currently a standard commercially available method to measure SGSH activity. In my opinion there is no need to provide the scheme of this reaction.
Discussion. MPS IIIA is a progressive disease. Did authors attempt to study survival of the KO adult fish and see if they present a shorter life span? Does hypoactive behavior aggravate with age? This needs to be discussed.
Author Response
Please find our reviewer comments attached. We thank Reviewer 3 for their in-depth, useful and helpful comments, which we have broadly implemented and will inform future work.

Round 2
Reviewer 2 Report
The authors have done a thorough response to reviewers however some of the justification has not been included in the revised manuscript. Specifically the following points needs to be included in the revised manuscript for additional clarity.
Comment #9 to Reviewer #2 has been addressed but has not been incorporated into the results (Section 2.4) or discussion of the revised manuscript. To aid readers not familiar with zebrafish models it would be useful to include comments on astrocyte reactivity and the absence of these in the zebrafish model. This is a significant disparity between this model and the established mouse models in which astrogliosis is thought to be a significant contributor to neuropathology. This should be included in text.
Also as reviewer 3 highlighted as well the microglia markers (Cd11b and Ilb4) and lack of conservation in zebrafish of theses typical mammalian markers would be useful to include in section 2.5 and the discussion and justification of use of morphology for this audience would be beneficial.
This (or an abridged version) should be included in the introduction and/or discussion to emphasise the aim of this paper to justify the use of early stage zebrafish upfront.
"The purpose of this paper was not to examine features of MPS IIIA pathobiology over the entire life-span or in senescent sgshΔex5-6 zebrafish. On the contrary, we are more interested in identifying early phenotypes and signs of pathobiology already detectable in embryonic or larval stages. The particular strength of the zebrafish model is use of embryos and larvae, and we think that our mutant model will be very useful for further studies on early aspects of MPS IIIA-like pathology and for in vivo testing of compounds."
Similarly for those not familar with the zebrafish, clarification on the timeline of zebrafish development and neurogenesis would be beneficial in the introduction. This is addressed in the comments to reviewers but not included in the manuscript in detail under comment #2 .
All other changes are relevant and I'm happy to accept for publication pending these additions to the manuscript.
Author Response
Please see attached our response to Reviewer 2's comments

Reviewer 3 Report
The authors have been responsive to reviewers' comments (with an exception of suggestion to improve the results of enzymatic SGSH activity assays by additional experimentation) and have strengthened what was already a good manuscript. Although I stand behind my opinion that improving assays and including proper blanks would strengthen the manuscript, I do recognize that even in their current form these experiments demonstrate the absence of SGSH activity in the tissues of knockout fish, which is probably sufficient for this first report. At the discretion of authors I would suggest them to acknowledge in the discussion that future analysis of SGSH activity and activities of other lysosomal enzymes in the fish tissues would be necessary to understand how the HS catabolic pathway functions in these species.
Author Response
Please find our response to Reviewer 3's comments attached
